# Why Warmup the Learning Rate? Underlying Mechanisms and Improvements

**Dayal Singh Kalra** [2]          **Maissam Barkeshli** [1]

## Abstract

It is common in deep learning to warm up the learning rate $\eta$, often by a linear schedule between $\eta_{\text{init}} = 0$ and a predetermined target $\eta_{\text{trgt}}$. In this paper, we show through systematic experiments using SGD and Adam that the overwhelming benefit of warmup arises from allowing the network to tolerate larger $\eta_{\text{trgt}}$ by forcing the network to more well-conditioned areas of the loss landscape. The ability to handle larger $\eta_{\text{trgt}}$ makes hyperparameter tuning more robust while improving the final performance. We uncover different regimes of operation during the warmup period, depending on whether training starts off in a progressive sharpening or sharpness reduction phase, which in turn depends on the initialization and parameterization. Using these insights, we show how $\eta_{\text{init}}$ can be properly chosen by utilizing the loss catapult mechanism, which saves on the number of warmup steps, in some cases completely eliminating the need for warmup. We also suggest an initialization for the variance in Adam which provides benefits similar to warmup.

## 1 Introduction

One of the most important choices to make in gradient-based optimization is the learning rate (step size) $\eta$. If $\eta$ is too small, then learning may take place too slowly or the model might get stuck in unfavorable regions of the loss landscape. If $\eta$ is too large, training will typically diverge. In practice, it is common to pick a dynamical learning rate schedule $\eta_t$ [2, 4, 40, 26]. Modern learning rate schedules for deep learning typically consist of a warmup period where $\eta_t$ is increased linearly from zero to a target value $\eta_{\text{trgt}}$ over a warmup time $T_{\text{wrm}}$ [13, 34]. After the warmup period, it is common to eventually decay the learning rate, for example via a cosine decay schedule [34, 26, 40].

Given that warmup is standard in the practitioner's toolkit, it is important to understand it deeply and identify improvements. In modern settings, perhaps the earliest work to use warmup was [14], which used a small constant learning rate for the first few epochs of training and then switched to a larger learning rate. A linear warmup schedule was later introduced in [13]. The intuition given was that to scale the minibatch size in SGD by a factor of $k$, it is natural to also scale the learning rate by a factor of $k$, provided the model is not changing too rapidly and successive gradients are roughly aligned. However at the beginning of training, the model is changing rapidly, so it is natural to start with a lower learning rate and gradually increase it to the target value after the network has stabilized.

Other explanations suggest that since the network is initialized randomly, the gradient steps at the beginning of training are not meaningful, and thus it would be harmful to take large steps in such directions [40], so it makes sense to take smaller steps early in training. The analysis by [12] suggests that warmup primarily limits the magnitude of weight updates in the deeper layers, preventing large instabilities. It has also been suggested that the key benefit of warmup arises for adaptive optimizers, such as Adam: [23] argues that the variance of the adaptive learning rate is large during early training because the network has seen too few training samples; it is asserted that this large variance is harmful,

---

[1]Department of Physics and Joint Quantum Institute, University of Maryland, College Park

[2]Institute for Physical Science and Technology, University of Maryland, College Park

38th Conference on Neural Information Processing Systems (NeurIPS 2024).

and that warmup acts as a variance reduction method by allowing the network to collect accurate statistics of the gradient moments before using larger learning rates. Alternatively, it is also sometimes stated that the initialization may start the model off at places in parameter space that are unstable, difficult to optimize, and easily lead to divergence, and that warmup can help alleviate this [40].

The above explanations are varied and do not clearly demonstrate why and to what extent warmup is necessary. A loss landscape perspective was given in [10] (and summarized in [26] Ch. 8), which argued that an important effect of warmup is to gradually reduce the sharpness (the top eigenvalue of the Hessian of the loss), thus causing the model to leave poorly conditioned areas of the loss landscape and move towards flatter regions which can tolerate larger learning rates. They argue that the mechanism for this is similar to the dynamical stability (catapult) mechanisms studied in [35, 22].

**Our contributions.**   Here we perform extensive studies on the effect of learning rate warmup across architectures (FCNs, ResNets, and Transformers), initializations and parameterizations, datasets (CIFAR-10, CIFAR-100, TinyImageNet, WikiText), and for both SGD and Adam.

We demonstrate through systematic experiments that by far the primary benefit of learning rate warmup is to allow the network to tolerate larger learning rates than it otherwise would have. This builds on the observations of [10] by showing that any other benefits are marginal, disentangling the effect of warmup duration and target learning rate, and by extending the empirical evidence to include adaptive optimizers and Transformers.

For SGD, the maximal allowable learning rate is determined by the sharpness (the top eigenvalue of the Hessian of the loss). As we discuss in Section 4, we find that there are several qualitatively distinct regimes and mechanisms at play. These depend on whether the network starts off in a sharpness reduction or progressive sharpening phase [18, 19, 5], which in turn depends on the initialization and parameterization. We further find that the performance of the network is largely determined by the target learning rate. For a fixed target learning rate, increasing the warmup time provides only marginal benefit, which arises by keeping the network further away from the divergence (failure) boundary. The ability of the network to withstand a larger target learning rate in turn makes hyperparameter tuning of the target learning rate more robust, since the network responds well to a larger window of target learning rates, possibly explaining the popularity of warmup.

We then investigate Adam in detail, and show that the underlying mechanisms of warmup are similar to the SGD case, but with sharpness replaced by a preconditioned sharpness (the top eigenvalue of the pre-conditioned Hessian, defined below) . Our results disagree somewhat with prior results [23] on the underlying reason for warmup's benefits: We find that the key issue is not observing too few training samples, but rather that the pre-conditioned sharpness typically starts off at high values (even in the large batch case), causing considerable instabilities at high learning rates. Such instabilities, which may be retained in Adam's memory, can result in performance degradation and even training failures. Warmup mitigates such instabilities by gradually pushing down the preconditioned sharpness, enhancing performance, and preventing training failures. We propose a simple alternative initialization for Adam, which we refer to as GI-Adam, which provides benefits similar to warmup and consistently improves over standard Adam by inducing lower preconditioned sharpness at initialization, thus pushing the training failure boundary to higher target learning rates. This also demonstrates a different way to remove the bias correction of RMSProp with momentum.

Our analysis shows how much of the time spent during the warmup period is wasted. We show that this wasted time can be saved by making use of the catapult mechanism [22] to effectively estimate the initial sharpness scale by line search, providing a more principled choice of $\eta_{\text{init}}$. Our experiments show that, depending on the target learning rate and initial sharpness, one can dramatically reduce the warmup time, and in some cases remove it altogether.

## 2   Notations and Preliminaries

**SGD(-M):** Given gradients $\boldsymbol{g}_t := \nabla_\theta L(\boldsymbol{\theta}_t)$ at step $t$, Stochastic Gradient Descent with momentum updates the parameters $\boldsymbol{\theta}_t$ using learning rate $\eta_t$ and momentum $\boldsymbol{m}_t$ with coefficient $\beta$. The update equations are: $\boldsymbol{m}_{t+1} = \boldsymbol{g}_t + \beta \boldsymbol{m}_t$ and $\boldsymbol{\theta}_{t+1} = \boldsymbol{\theta}_t - \eta_t \boldsymbol{m}_{t+1}$. $\beta = 0$ corresponds to SGD.

**Adam:** Adam [20] updates the parameters $\boldsymbol{\theta}_t$ according to the equations: $\boldsymbol{m}_{t+1} = \beta_1 \boldsymbol{m}_t + (1-\beta_1)\boldsymbol{g}_t$, $\boldsymbol{v}_{t+1} = \beta_2 \boldsymbol{v}_t + (1-\beta_2)\boldsymbol{g}_t^2$, and $\boldsymbol{\theta}_{t+1} = \boldsymbol{\theta}_t - \eta_t \frac{\hat{\boldsymbol{m}}_{t+1}}{\sqrt{\hat{\boldsymbol{v}}_{t+1}}+\epsilon} = \boldsymbol{\theta}_t - \eta_t P_{t+1} \boldsymbol{m}_{t+1}$, where $\hat{\boldsymbol{m}}_t = \frac{\boldsymbol{m}_t}{1-\beta_1^t}$

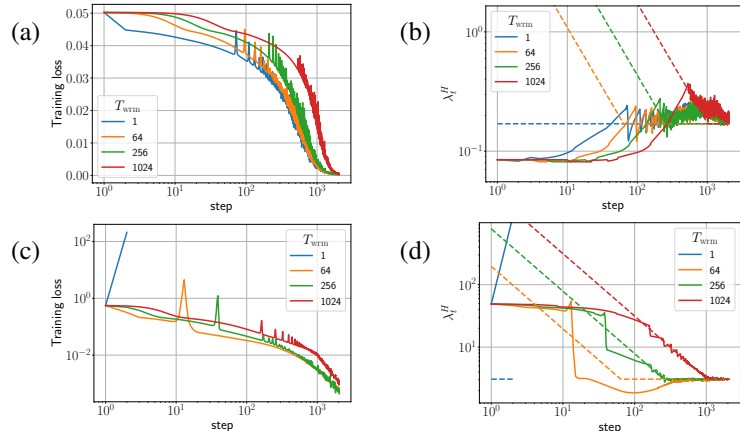

Figure 1: Training loss and sharpness trajectories of FCNs trained on a 5k subset of CIFAR-10 with MSE loss using GD. In the sharpness plot, the dashed lines represent the $2/\eta_t$ curves, and when $\lambda_t^H$ is above these curves, training exceeds the instability threshold ($\eta > \eta_c$). (top) $\mu$P with $\eta_{\text{trgt}} = 1/\lambda_0^H$, (bottom) SP with $\eta_{\text{trgt}} = 32/\lambda_0^H$. Similar mechanisms are observed across different architectures, loss functions, and mini-batch sizes, as shown in Appendix E.

and $\hat{\boldsymbol{v}}_t = \frac{\boldsymbol{v}_t}{1-\beta_2^t}$ are the bias-corrected moments, $\epsilon$ is a small scalar used for numerical stability and $P_t = (1 - \beta_1^t)\left[\text{diag}\left(\sqrt{\hat{\boldsymbol{v}}_t}\right) + \epsilon\mathbf{I}\right]$ is the preconditioner.

**Linear Warmup:** This is defined by the schedule $\eta_t = \eta_{\text{init}} + (\eta_{\text{trgt}} - \eta_{\text{init}})t/T_{\text{wrm}}$. The warmup rate is $\alpha := (\eta_{\text{trgt}} - \eta_{\text{init}})/T_{\text{wrm}}$. $T_{\text{wrm}} = 1$ corresponds to constant learning rate. Unless otherwise specified, we set $\eta_{\text{init}} = 0$ when referring to linear warmup. We propose strategies for selecting $\eta_{\text{init}}$ in Section 6.

**Sharpness:** The sharpness is defined as the maximum eigenvalue of the Hessian of the loss $\lambda_t^H := \lambda_{\max}(\nabla_\theta^2 L)$, with subscript $t$ indexing the training step. Adaptive optimizers, such as Adam, effectively perform gradient descent in a transformed space determined by $\phi := P^{1/2}\theta$ (for details, see Appendix B.1). Hence, their stability is determined by the largest eigenvalue of the pre-conditioned Hessian, denoted by $\lambda^{P^{-1}H} := \lambda_{\max}(P^{-1}\nabla_\theta^2 L)$, rather than the sharpness itself.

**Parameterizations in Neural Networks:** The mechanism of warmup and its effectiveness is heavily influenced by the network parameterization (see Sections 4 and 5). Standard Parameterization (SP) [33] is a staple in common libraries [28, 3]. Another notable parameterization is the Neural Tangent Parameterization (NTP) [17], which along with SP resides in the kernel learning class at infinite width. Ref. [37] proposed Maximal Update Parameterization ($\mu$P) which exhibits feature learning at infinite width. Neural network parameterizations significantly impact training dynamics [19].

## 3 Overview of Training Instabilities and the Self-Stabilization Mechanism

One important underlying mechanism of warmup is intimately tied to training instabilities. These training instabilities, often referred to as 'catapults' [22, 5], arise when the learning rate $\eta$ exceeds a critical threshold $\eta_c$, where both $\eta$ and $\eta_c$ generally change with time. The critical learning rate $\eta_c$ is influenced by a variety of factors, including the choice of optimizer [5, 6], mini-batch size [35, 6], and model properties such as depth, width, parameterization, and initialization [18, 19]. For a detailed overview of instability thresholds, see Appendix B.

When the instability threshold is exceeded ($\eta > \eta_c$), two cases arise: (i) if the learning rate is higher than the instability threshold but smaller than a maximum stable learning rate (which varies with time), i.e., $\eta_c < \eta < \eta_{\max}$, training stabilizes through a self-stabilization process and training continues, (ii) if the learning rate exceeds this maximum stable learning rate $\eta > \eta_{\max}$, training experiences severe instabilities. For SGD, these can result in training divergence, characterized by the loss increasing to infinity. For Adam, training may cease, resulting in a training failure, where the loss fails to improve significantly over its initial value, as we demonstrate in Section 5.

The self-stabilization mechanism of GD can be understood through both empirical observations (Figure 1) and a theoretical model. We first describe a model derived by Ref. [7] that effectively captures this phenomenon. The model assumes that the top eigenvector $\boldsymbol{u}$ changes slowly through

training and can be treated as constant and considers a cubic approximation of the GD dynamics around a reference point $\boldsymbol{\theta}^*$. The dynamics along the projection $x_t := \boldsymbol{u}^T(\boldsymbol{\theta}_t - \boldsymbol{\theta}^*)$ is given by two coupled non-linear equations:

$$x_{t+1} = (1 - \eta_t \lambda_t^H)x_t, \qquad \lambda_{t+1}^H = \lambda_t^H + \eta_t(\alpha - \beta x_t^2), \tag{1}$$

where $\alpha := -\nabla \lambda^H \cdot \nabla L$ quantifies the instantaneous change in sharpness and $\beta := \|\nabla \lambda^H\|^2$ controls the non-linear change in sharpness. In this model, an instability arises when $\eta_t > \eta_c = 2/\lambda_t^H$. Ref. [7] considered a constant learning rate $\eta$ and assumed progressive sharpening ($\alpha > 0$). In contrast, we consider a time-dependent learning rate and allow $\alpha$ to attain both positive and negative values in order to analyze different warmup mechanisms.

The self-stabilization mechanism manifests as a four-step process [22, 7]. Below, we describe the four steps of the self-stabilization mechanism using the above model and the $T_{\mathrm{wrm}} = 64$ trajectories illustrated in Figure 1(c, d):

(1) **Approaching instability:** Due to increasing learning rate and/or progressive sharpening, training approaches the instability threshold $\eta_t = \eta_c = 2/\lambda_t^H$. In Figure 1(d), this occurs within the first 10 steps due to increasing learning rate.

(2) **Blow up:** On exceeding the instability threshold ($\eta > \eta_c$), Equation (1) predicts exponential growth in $x_t$, empirically manifesting as a sharp increase in loss, as observed in Figure 1(c).

(3) **Sharpness reduction:** For small enough learning rates, $|x_t|$ (and the loss) continues to increase until the higher-order term in the sharpness update equation causes a decrease in sharpness ($x_t > \alpha/\beta$). This is observed as an abrupt decrease in sharpness in Figure 1(d). If the sharpness fails to decrease over extended steps, it may result in training divergence (e.g., see $T_{\mathrm{wrm}} = 1$ case in the same figure).

(4) **Return to stability:** Once the sharpness has decreased appreciably so that $\eta_t \lambda_t^H < 2$, stability is restored and the loss eventually decreases.

While the self-stabilization process for more complex optimizers remains poorly understood, a qualitatively similar mechanism is observed in practice, as we will see in the later sections.

## 4 Warmup Mechanisms of Gradient and Adaptive Methods

This section analyzes the underlying mechanism of warmup through the lens of sharpness dynamics. A key finding is that warmup decreases sharpness in two ways: by allowing a natural sharpness reduction effect at early times and/or by forcing sharpness reduction through training instability at later times.

### 4.1 Stochastic Gradient Descent

Learning rate warmup is intrinsically tied to sharpness dynamics, as sharpness determines the instability threshold $\eta_c$. As the learning rate is increased during warmup, training instabilities can be triggered. Assuming the warmup rate is not too high, these instabilities induce a temporary increase in the loss and a decrease in the sharpness to restore stability through the self-stabilization mechanism. Ultimately this allows the model to adapt to the increased learning rate. In other words, a primary goal of warmup is to gradually reduce sharpness, guiding training towards flatter regions that can accommodate training at higher learning rates [10].

However, digging deeper, we find that training has a 'natural' preference for sharpness evolution throughout the training course [19]. Before exceeding the instability threshold ($\eta < \eta_c$), training naturally experiences either a progressive increase or decrease in sharpness, as observed in Figure 1, which is unrelated to warmup. For instance, consider the sharpness trajectories with $T_{\mathrm{wrm}} = 1024$ in the above figure. In Figure 1(b), sharpness has a natural preference for increasing, whereas in Figure 1(d), it tends to decrease on its own. This natural sharpness evolution can defined as the sharpness evolution under gradient flow, corresponding to $\alpha$ in Equation (1). The interplay between this natural sharpness evolution and the deliberate intervention of warmup to reduce sharpness can result in completely distinct dynamics. Below, we use the model described by Equation (1) and the experiments in Figure 1 to describe these distinct dynamics.

**(C1) Natural Progressive Sharpening** ($\alpha > 0$; top row of Figure 1): The combined effect of naturally increasing sharpness while the learning rate is also being increased results in a "head-on

collision" at which the instability threshold is exceeded ($\eta_t > \eta_c$). This causes the loss to increase, leading to a decrease in sharpness. Once the sharpness has decreased appreciably, the stability is restored ($\eta_t < \eta_c$). As training proceeds, both sharpness and learning rate continue to increase, again surpassing the instability threshold. This results in a *persistent catapult cycle*, characterized by $\eta_t \approx {}^2/\lambda_t^H \approx \eta_c$, for the remainder of the warmup period, as seen in Figure 1(b).

**(C2) Natural Sharpness Reduction** ($\alpha < 0$; bottom row of Figure 1): The network is naturally already reducing its sharpness during early training. However, if the learning rate is increased sufficiently quickly, eventually the instability threshold will be reached (akin to a "rear-end collision"), causing the loss to increase. For small enough learning rates, the increased loss induces a dramatically more pronounced decrease in sharpness than would naturally occur, ultimately restoring stability ($\eta_t < \eta_c$). To exceed the instability threshold again, the learning rate must significantly increase to account for the decreased sharpness, potentially requiring considerable training steps. Consequently, training experiences one or more separated catapults during the warmup phase, as seen in Figure 1(c, d). This contrasts with the progressive sharpening case, where training enters a continuous catapult cycle after reaching the instability threshold for the first time. Notably, training may eventually reach a very flat region of the landscape during warmup, with gradients pointing towards increasing sharpness (e.g., $T_{\text{wrm}} = 64$ in Figure 1(d)). Upon reaching such a region, the dynamics aligns with the natural progressive sharpening scenario.

When natural sharpness reduction is significant (large negative $\alpha$), warmup may not need to actively reduce sharpness. Instead, it may "piggy-back" on the inherent sharpness decrease, resulting in a completely different warmup mechanism, which does not rely on instabilities to facilitate training at higher learning rates.

The above two scenarios can be interpreted as cooperative or competitive dynamics between warmup and the natural evolution of sharpness. When training inherently undergoes sharpness reduction, it cooperates with warmup in decreasing sharpness. Conversely, if the natural trajectory of training is towards increasing sharpness, it opposes the warmup's effort, leading to a persistent cycle of catapults.

**The Effect of Warmup Duration:** Given a fixed target learning rate $\eta_{\text{trgt}}$, increasing the warmup duration $T_{\text{wrm}}$ delays the point at which training exceeds the instability threshold $\eta_c$, allowing the sharpness to evolve freely before reaching this point. In the sharpness reduction case, sharpness can significantly decrease by the time this threshold is reached, lowering the need for warmup to decrease sharpness actively. Consequently, increasing $T_{\text{wrm}}$ results in catapults that are both delayed and smaller in magnitude, as seen in Figure 1(d). As the catapults become less intense on increasing the warmup duration, the model can train at higher target learning rates without diverging. For extended warmup durations, warmup may not actively reduce sharpness in these sharpness reduction cases and instead it leverages the inherent sharpness decrease.

In the progressive sharpening case, increasing $T_{\text{wrm}}$ allows the sharpness to naturally increase. As a result, training exceeds the instability threshold for the first time at a relatively lower learning rate compared to the constant learning rate case. Although warmup has to now undertake more work in decreasing sharpness, it does so in a more gradual manner since increasing the warmup duration amounts to a lower warmup rate $\eta_{\text{trgt}}/T_{\text{wrm}}$. As a result, the fluctuations observed on exceeding the instability threshold are much smaller in magnitude, as seen in Figure 1(a, b).

**Small vs. Large Initializations:** So far, we have outlined different warmup mechanisms without describing specific conditions that typically exhibit them. Small initializations, such as those using maximal update parameterization ($\mu$P) [37] in the large width limit or appropriately using normalizing layers (e.g. standard Transformer architectures, see Figure 17 in Appendix E.5), are characterized by a small initial network output. Such initializations start in flat regions where gradients point toward increasing sharpness [19], placing them in the progressive sharpening category (C1). As we will see in Section 5, such initializations may not significantly benefit from warmup as they already start in a flat region. In contrast, large initializations, such as FCNS, CNNs, ResNets with Standard Parameterization (SP) initialized at criticality [29, 31] or Transformers with the last layer-norm removed, undergo an early sharpness reduction, categorizing them into sharpness reduction category (C2). As the primary effect of warmup is to reduce sharpness, we expect such large initializations to considerably benefit from warmup. Notably, large initializations can eventually undergo progressive sharpening at later training stages [18, 19] and adhere to the second mechanism, especially for prolonged warmups.

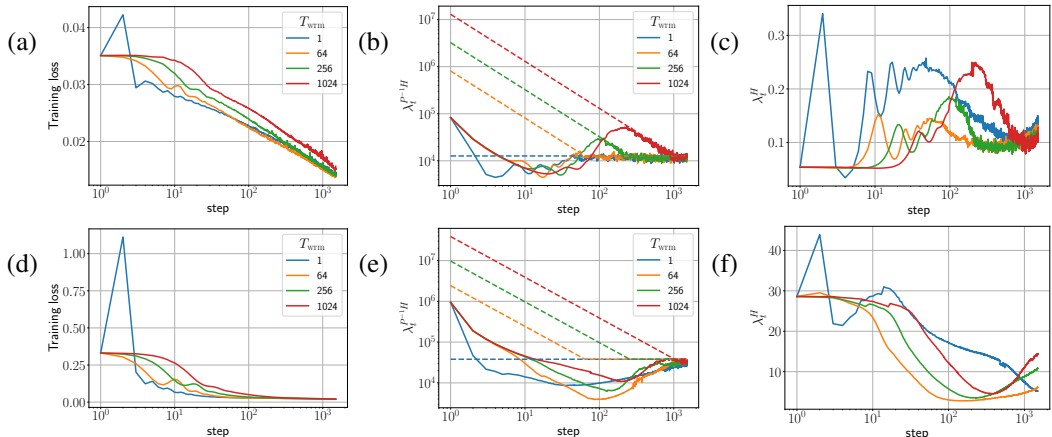

Figure 2: Training loss and sharpness trajectories of FCNs trained on the entire CIFAR-10 dataset with MSE loss using full batch Adam. (top) simple-$\mu$P (for details, see Appendix D.2.1) with $\eta_{\text{trgt}} = 0.003$ and (bottom) SP with learning rate $\eta_{\text{trgt}} = 0.001$. The dashed lines in the sharpness figures illustrate the instability thresholds $(2+2\beta_1)/\eta_t(1-\beta_1)$. Similar mechanisms are observed for different architectures, loss functions, and smaller batch sizes as detailed in Appendix E.

Natural sharpness change provides an intuitive way of determining whether an initialization is 'small' or 'large': if training from a given initialization exhibits sharpness reduction, it suggests the existence of naturally flatter initializations that could be chosen instead. This observation is particularly helpful for analyzing Adam in the later sections and motivates modifications to improve it.

## 4.2 Stochastic Gradient Descent with Momentum (SGD-M)

The warmup mechanism of SGD-M, while at its core is similar to that of vanilla SGD, has a few subtleties. Here we summarize the major differences, leaving details to Appendix E.2.

During early training, the loss may decrease non-monotonically on incorporating momentum, even at small learning rates. Such oscillations are also observed when quadratic loss functions are optimized using GD with momentum [11]. These oscillations make it challenging to differentiate between warmup-induced catapults and fluctuations in loss due to the intrinsic effects of momentum. Nevertheless, we can still observe loss spikes correlated with an abrupt decrease in sharpness at large learning rates, as detailed in Appendix E.2.

Additionally, the instability threshold $\eta_c$ itself evolves differently during training. It changes from $2/\lambda_0^H$ at initialization to $(2+2\beta)/\lambda_t^H$ later in training. Moreover, the late-time instability threshold is significantly influenced by the batch size, exhibiting a much smaller value than SGD for the same batch size. These properties make it more challenging to analyze the training dynamics of SGD with momentum. Nonetheless, the fundamental warmup mechanisms closely mirror the vanilla SGD case. We leave a more detailed analysis of the early training dynamics of SGD-M for future studies.

## 4.3 Adaptive Gradient Methods (Adam)

Adaptive optimizers effectively perform gradient descent in a transformed space given by $\phi = P^{1/2}\theta$, as we show in Appendix B.1. This analysis suggests that their local stability should be determined by the largest eigenvalue of $\nabla_\phi^2 L = P^{-1}\nabla_\theta^2 L$, which we refer to as the pre-conditioned Hessian. Indeed, this is what we observe in Figure 2, which shows the training loss, pre-conditioned sharpness $\lambda^{P^{-1}H}$, and sharpness trajectories for full batch Adam. In these figures, sharpness is significantly smaller than its instability threshold $(2+2\beta_1)/\eta_t \approx 4000$, indicating that sharpness does not determine stability. Instead, loss catapults are associated with $\lambda^{P^{-1}H}$ exceeding its instability threshold.

The pre-conditioned sharpness starts high for both progressive sharpening (simple-$\mu$P) and sharpness reduction (SP) scenarios considered in the previous section. For simplicity, we considered a simpler version of $\mu$P, detailed in Appendix D.2.1. In particular, for $\mu$P models, $\lambda_0^{P^{-1}H} \sim 10^5$ despite being initialized in a flat region as measured by sharpness, while for SP models, $\lambda_0^{P^{-1}H} \sim 10^6$. These large initial values arise because $P_1 = (1-\beta_1)\left[\text{diag}(\boldsymbol{g}_0^2) + \epsilon\mathbf{I}\right]$ and $\boldsymbol{g}_0$ has components that are near zero.

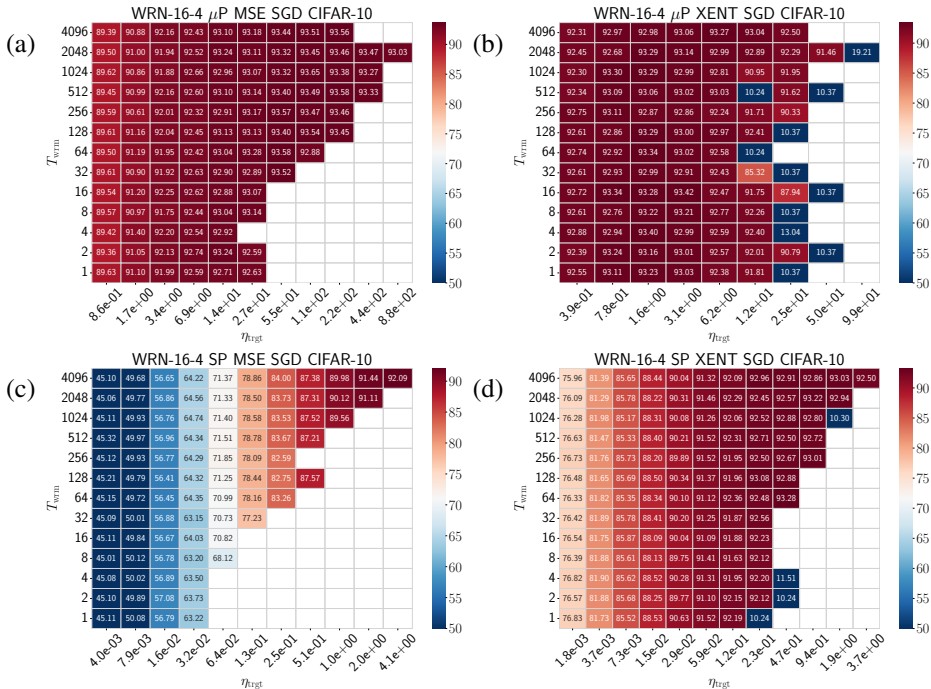

Figure 3: Test accuracy heatmaps of WRNs trained on CIFAR-10 using different and parameterizations loss functions using SGD: (a) $\mu$P and MSE loss, (b) $\mu$P and cross-entropy loss, (c) SP and MSE loss, and (d) SP and cross-entropy loss. Empty cells correspond to training divergences. Similar phase diagrams are generically observed for different architectures and datasets, as shown in Appendix F.

The large $\lambda_0^{P^{-1}H}$ can lead to training failures if the learning rate does not start sufficiently small. We put forward modifications to improve Adam in Section 6; here we continue characterizing the warmup mechanisms of Adam.

Given that the pre-conditioned sharpness consistently starts high and decreases during early training, this behavior can be viewed as an extreme example of the natural sharpness reduction scenario (C2) described in the previous section. Training Adam at high initial learning rates without warmup can cause large catapults, as seen in Figure 2(d), potentially leading to training failures. Increasing the warmup duration allows the pre-conditioned sharpness to naturally decrease. This prevents the loss from spiking during early training and avoids training failures. In the later stages of training, the pre-conditioned sharpness may continue reducing or exhibit progressive sharpening. From here on, the dynamics follows the warmup mechanisms discussed in the previous sections, with sharpness replaced with pre-conditioned sharpness. Similar to the momentum case, Adam's stability threshold at late training times significantly decreases for smaller batch sizes [6], also shown in Appendix E.4.

## 5 Impact of Warmup on Training and Generalization

Here we investigate the impact of warmup on training efficacy and generalization by disentangling the role of $\eta_{\text{trgt}}$ and $T_{\text{wrm}}$. Our key findings are that generalization capability is primarily determined by $\eta_{\text{trgt}}$ and that Adam is particularly sensitive to large catapults. The role of increasing $T_{\text{wrm}}$ is to (i) allow the network to tolerate larger $\eta_{\text{trgt}}$, and (ii) move training further away from the divergence (failure) boundary, leading to a marginal improvement in generalization.

**Experimental Setup:** We consider WideResNets (WRNs) and Transformers (LM) parameterized in either SP or $\mu$P. WRNs are trained on CIFAR-10, CIFAR-100, and Tiny-ImageNet, employing data augmentation. Transformers are trained on the next token prediction task using the WikiText dataset. These models are trained with MSE or cross-entropy (xent) loss functions using SGD or Adam optimizers for a fixed training budget of $T = 10^5$ steps unless otherwise specified. Training begins with a linear warmup phase from $\eta_{\text{init}} = 0$ to $\eta_{\text{trgt}}$ over $T_{\text{wrm}}$ steps. After warmup, training continues at $\eta_{\text{trgt}}$ for the remaining training budget. In some cases, following the warmup period, we decrease the learning rate using cosine decay [24]. Further details are provided in Appendix D.

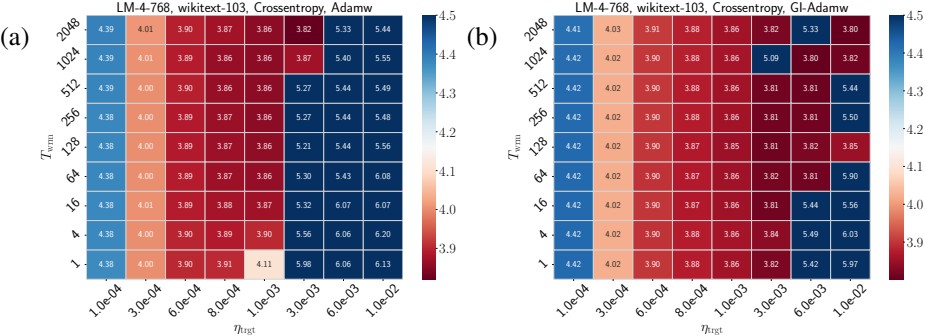

Figure 4: Test loss heatmaps of Pre-LN Transformers in SP trained on WikiText-103 with cross-entropy loss for a single epoch using (a) Adam, and (b) GI-Adam (introduced in Section 6). Additional results are presented in Appendix F.3.

## 5.1 Stochastic Gradient Descent (SGD)

Figure 3 presents heatmaps that show the best test accuracy achieved during training, plotted in the $\eta_{\text{trgt}}$-$T_{\text{wrm}}$ plane for different parameterizations and loss functions. These phase diagrams of warmup also show the convergence-divergence boundary, with empty cells indicating training divergences, illustrating the interplay between warmup duration and the maximum trainable $\eta_{\text{trgt}}$. Below, we discuss the crucial insights these results provide into warmup's role in training dynamics.

**Longer Warmup Facilitates Training at Higher Learning Rates:** These phase diagrams reveal that an extended warmup duration facilitates training at higher target learning rates. This benefit is particularly noticeable for large initializations (like SP) and MSE loss. In contrast, the advantage is less pronounced when using cross-entropy loss and smaller initializations (like $\mu$P). The diminished benefit for $\mu$P is likely due to its initialization in a relatively flat region of the loss landscape, which can already facilitate training at higher learning rates at initialization. This consistent increase in maximum $\eta_{\text{trgt}}$ with warmup durations can be understood through the lens of warmup mechanisms described in the previous section. As observed in Figure 1, when the warmup duration is increased, loss catapults occurring on surpassing the instability thresholds become milder. This effectively pushes the divergent boundary to higher learning rates.

**Final Performance Primarily Depends on the Target Learning Rate:** A closer look into these phase diagrams reveals that, slightly away from the divergent boundary, the test accuracy primarily depends on the target learning rate and nominally on the warmup duration. Based on the model performance, we can categorize these phase diagrams into two distinct cases: (i) models that fail to achieve optimal performance when trained with a constant learning rate (e.g., Figure 3(c)), and (ii) models that attain optimal performance without warmup (e.g., Figure 3(b)). The first scenario corresponds to models with large initializations. Increasing the warmup duration improves performance by facilitating training at higher learning rates. Yet, similar performance is observed for different warmup durations, suggesting that the primary gain comes from the target learning rate, rather than the duration itself. The second case arises for flat initializations, which can already train at large learning rates, and resultantly the optimal performance is already achieved without warmup. While increasing warmup duration facilitates training at even higher learning rates, it does not enhance performance. Nevertheless, it does broaden the range of optimal learning rates, reducing the need for precise tuning of the target learning rate, and making training more practical and robust. We conclude that warmup can serve two key purposes: (i) it can significantly improve model performance in large initialization cases, and (ii) extend the range of optimal target learning rates for small initializations, making it easier to tune the target learning rate. In Appendix F.2, we demonstrate that these results hold on incorporating momentum and employing cosine learning rate decay.

## 5.2 Adaptive Gradient Methods (Adam)

Figure 4(a) shows the warmup phase diagram of Adam. Increasing the warmup duration enables training at higher learning rates by allowing the pre-conditioned sharpness to decrease naturally, thereby reducing the severity of catapults. These large catapults, which may persist in Adam's memory, can lead to performance degradation and training failures. Thus, in addition to facilitating

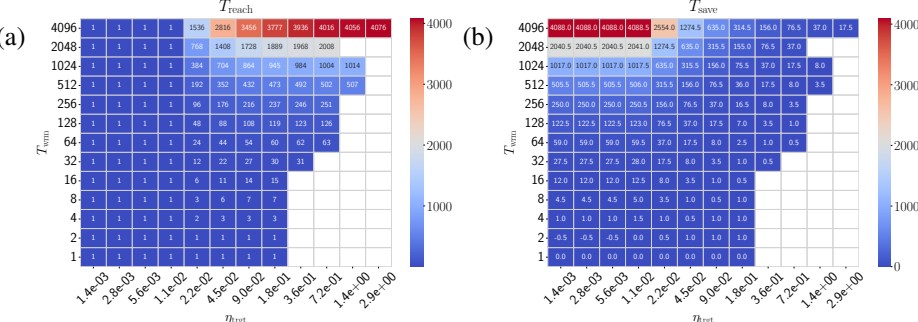

Figure 5: Heatmaps showing (a) the steps to reach $\eta_{\text{trgt}}$ ($T_{\text{reach}}$) and (b) the effective steps saved ($T_{\text{save}}$) on setting $\eta_{\text{init}} = \eta_c$ for WRNs in SP trained on CIFAR-10 using SGD with cross-entropy loss.

training at higher rates similar to SGD, warmup further improves Adam's performance by addressing its vulnerability to large catapults, justifying its widespread use with Adam. Below, we discuss the distinct properties of Adam phase diagrams in detail.

**Training Failures of Adam:** Remarkably, we find that models trained with Adam always exhibit training failures rather than divergences where the loss grows without bound, as further demonstrated in Appendix G. In cases of training failure, we often observed that certain layers or residual blocks output zero, leading to vanishing gradients. This implies that the model gets stuck at a critical point and is unable to train further. Understanding this unexpected phenomenon requires further study.

**Performance Degradation prior to Failure Boundary:** Test accuracy in these phase diagrams declines well before the failure boundary, in contrast to SGD where optimal learning rates are observed near the divergence boundary. This discrepancy stems from Adam's property of retaining a memory of gradient magnitudes. At large learning rates, along with the loss, the gradients spike during early training, as seen in Figure 28 in Appendix G. While the gradients decrease after a few steps, the second moment $v$ remains large for an extended period, leading to a small effective learning rate $\eta P^{-1}$. As a result, training struggles to escape high-loss regions. Therefore, a longer warmup is more beneficial for Adam compared to SGD, as it is crucial to stay away from the failure boundary.

## 6 Improved Hyperparameter Initialization Schemes for Optimizers

**Initial Learning Rate Selection for Warmup:** Setting the initial learning rate to $\eta_{\text{init}} = 0$ is common practice in warmup [27, 8]. Our analysis reveals that the primary effect of warmup is to facilitate training at higher learning rates by annealing sharpness (or pre-conditioned sharpness for Adam). From this perspective, starting with $\eta_{\text{init}} = 0$ appears suboptimal, as it can significantly delay the learning rate from exceeding the instability threshold, thus delaying the primary effect of warmup.

An effective strategy involves setting $\eta_{\text{init}} = \eta_c$ to induce loss increase and thereby sharpness decrease right from initialization. We introduce a straightforward search method that only uses forward passes to estimate the initial critical learning rate $\eta_c$. The method consists of two stages: (i) an exponential search, starting from an initial guess $\eta_0$, iteratively multiplies $\eta_0$ by a factor $k > 1$ until the loss increases. This identifies an interval $[\eta_{\text{lwr}}, \eta_{\text{uppr}}]$ containing $\eta_c$, (ii) a binary search further narrows down $[\eta_{\text{lwr}}, \eta_{\text{uppr}}]$ by evaluating the loss at the midpoint $\eta_{\text{mid}} = (\eta_{\text{lwr}} + \eta_{\text{uppr}})/2$. If the loss increases, $\eta_{\text{uppr}}$ is updated to $\eta_{\text{mid}}$; otherwise, $\eta_{\text{lwr}}$ is set to $\eta_{\text{mid}}$. This process is repeated until the loss in the next step $L(\theta_1)$ satisfies the condition $L(\theta_1) < L(\theta_0)(1 + \delta)$, for some hyperparameter $\delta > 0$. For details, see Appendix B.3.

By setting $\eta_{\text{init}} = \eta_c$, training can achieve the target learning rate earlier. Consider the modified warmup schedule: $\eta_t = \eta_{\text{init}} + \eta_{\text{trgt}}(t/T_{\text{wrm}})$, which attains $\eta_{\text{trgt}}$ in $T_{\text{reach}} = T_{\text{wrm}}(1 - \eta_c/\eta_{\text{trgt}})$ steps, saving $T_{\text{wrm}}(\eta_c/\eta_{\text{trgt}})$ steps. Incorporating the computational cost of additional forward passes $T_{\text{fp}}$ required for estimating $\eta_c$ ($\sim 10$ in number), and noting that one training step approximately equates to two forward passes, the net computational savings is $T_{\text{save}} = T_{\text{wrm}}(\eta_c/\eta_{\text{trgt}}) - T_{\text{fp}}/2$. Figure 5 demonstrates how $T_{\text{reach}}$ and $T_{\text{save}}$ vary with the $T_{\text{wrm}}$ and $\eta_{\text{trgt}}$. For $\eta_{\text{trgt}} < \eta_c$, the target learning rate is reached in a single step, nearly saving the entire duration of the warmup, whereas for ($\eta_{\text{trgt}} > \eta_c$), starting $\eta_{\text{init}} \gtrsim \eta_c$ can save up to half of the allocated warmup duration, although this saving diminishes on approaching the divergent/failure boundary.

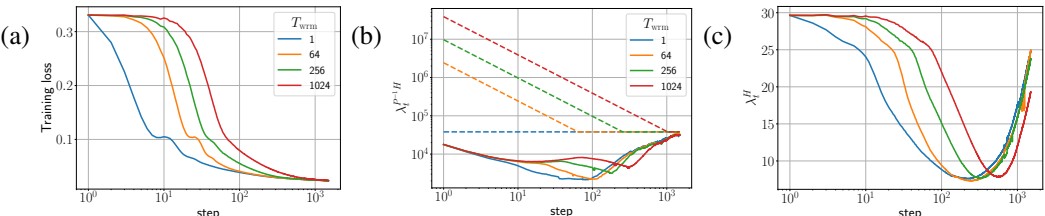

Figure 6: Training loss and sharpness trajectories of FCNs in SP. The experimental setup is identical to Figure 2 but with GI-Adam instead of standard Adam.

It is worth noting that there can be instances where there is no loss catapult at initialization. In our experiments, this only occurs for Transformers trained using SGD. In such scenarios, the prescribed approach is not applicable and one can resort to heuristics, such as setting the initial learning rate to a fraction of the maximum stable learning rate, such as $\eta_{\text{init}} = \eta_{\text{trgt}}/10$.

**GI-Adam: Improving Adam by Initializing The Second Moment using Gradients**

In Section 4.3, we observed that the pre-conditioned sharpness for Adam starts at a high value, even for low sharpness initializations like $\mu$P, and can lead to training failures at large learning rates. We propose Gradient Initialized Adam (GI-Adam), which initializes the second moment using the gradient squared, $v_0 = g_0^2$. In Appendix I.2, we show that a bias correction is not required when the second moment is initialized using the gradients. As a result, GI-Adam can be viewed as standard Adam with an automated warmup given by $\eta_t = \eta_{\text{trgt}} \sqrt{1 - \beta_2^t}$.

This simple trick reduces the initial pre-conditioned sharpness by around two orders of magnitude (more precisely by a factor of $\sqrt{1 - \beta_2}$) at initialization, preventing large catapults, as illustrated in Figure 6 (c.f. Figure 2(d-f)). Moreover, it consistently shows improvement over standard Adam across datasets and prevents training failures by pushing the training failure boundary to higher $\eta_{\text{trgt}}$, as shown in Figure 4(b). We provide additional results for different datasets in Appendix F.3. In Appendix F.4, we show that GI-Adam consistently performs on par or better than RAdam [23] while offering a simple modification to Adam.

To further assess that the primary cause of instability during early training is the large pre-conditioned sharpness, we randomly initialize $v_0$ but with the same norm as the gradients at initialization. Like GI-Adam, this also results in improved performance as shown in Appendix I.3.

We further reduce the pre-conditioner size by removing bias correction for the first moment, referred to as Flat-Adam. As demonstrated in Appendix I.4, this modification eliminates the initial decrease in pre-conditioned sharpness. We leave a comprehensive evaluation of Flat-Adam for future work.

## 7 Discussion

Our analysis provides new insights into the role of warmup across optimizers and parameterizations. We found compelling evidence that the primary effect of warmup is to facilitate training at higher learning rates and stabilizing the training dynamics by keeping it away from the failure (divergence) boundary. Looking under the hood, we found a variety of underlying mechanisms, which also suggested several improvements for hyperparameter initialization. In Appendix A, we provide practical guidance for practitioners on choosing the warmup duration.

Our analysis also motivates a potential parameter-free warmup strategy, which we refer to as *persistent catapult warmup*. The central idea behind this strategy is to repeatedly induce catapults aimed to progressively reduce sharpness, thereby facilitating training at higher learning rates. We present encouraging preliminary results in Appendix C and defer further development to future work.

The maximum learning rate can be written as $\eta_{\max} = c_{\max}/\lambda^{P^{-1}H}$. Here we showed how warmup effectively decreases $\lambda^{P^{-1}H}$, which is a local measure of sharpness. There is also another possible effect of warmup, that it can cause an increase in $c_{\max}$, which can be viewed as a more non-local measure of sharpness. Further analysis is required to understand how warmup helps increase $c_{\max}$.

**Limitations:** Our experiments were conducted on relatively small-scale datasets and models, and further investigations are needed to understand the generalizability of our findings to larger-scale settings. For Adam, we did not explore the dependence on hyperparameters $\beta_1$, $\beta_2$, $\epsilon$.

## Acknowledgments and Disclosure of Funding

We thank Tianyu He, Darshil Doshi, Andrey Gromov, Dan Roberts, Jeremy Cohen, Jonas Geiping, Yuqing Wang and Alex Damian for discussions and comments on the draft. The authors acknowledge the University of Maryland supercomputing resources (`http://hpcc.umd.edu`) made available for conducting the research reported in this paper. This work is supported in part by NSF DMR-2345644 (MB).

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

# A    Practical Guidance for Practitioners

**How to Select the Warmup Duration?**    Given a target learning rate $\eta_{\text{trgt}}$, if the training loss during the warmup period exhibits large instabilities (loss spikes), the warmup duration $T_{\text{wrm}}$ should be increased until such instabilities are sufficiently small. This effectively moves training away from the divergent / failure boundary, as illustrated in Figure 3. This is particularly crucial for Adam, as large instabilities can be detrimental and lead to considerable performance degradation without divergence, as discussed in Section 5.2.

**How to Select the Target Learning Rate?**    As the primary effect of warmup is to anneal sharpness by increasing the learning rate beyond the instability threshold, it suggests that the target learning rate should be at least greater than the instability threshold at initialization.

**When to Decay the Learning Rate?**    Figure 21 suggests that employing learning rate decay at small learning rates can result in performance degradation for a fixed training budget. Therefore, the learning rate should be decayed at large target learning rates only. The underlying intuition is that we use large target learning rates to train in a flat region of the landscape. However, these large learning rates restrict training to go into sharper regions of the basin and learning rate decay helps.

**Leveraging $\mu$P for Effecient Training:**    Our analysis suggests that the primary role of warmup facilitates training at higher learning rates by gradually reducing sharpness. Given this perspective, beginning training with flat initializations, such as $\mu$P, is advantageous. These initializations might allow for achieving optimal performance without the need for warmup, as observed in Figure 3.

# B    Instability Thresholds

## B.1    Instability Thresholds for Adaptive Optimizers

For adaptive optimizers, such as Adam, a pre-conditioner $P_{t+1}$ multiplies the gradients in the update equation

$$\boldsymbol{\theta}_{t+1} = \boldsymbol{\theta}_t - \eta_t P_{t+1}^{-1} \nabla_\theta L(\boldsymbol{\theta}_t). \tag{2}$$

Next, we define $\phi_t := P_{t+1}^{1/2}\boldsymbol{\theta}_t$, then the gradient and Hessian are $\nabla_\phi L = P^{-1/2}\nabla_\theta L$ and $\nabla_\phi^2 L := P^{-1}\nabla_\theta^2 L$. As a result, the update equations for $\phi$ is given by

$$\phi_{t+1} = \phi_t - \eta_t \nabla_\phi L(\boldsymbol{\theta}_t). \tag{3}$$

We observe that the update equation of an adaptive optimizer mirrors that of SGD under a change of variables. Using the stability arguments of SGD, we conclude that $\lambda_{\max}(P_{t+1}^{-1}\nabla_\theta^2 L(\boldsymbol{\theta}_t))$ determines the stability.

## B.2    Overview of Instability Thresholds

Lewkowycz et al. [22] showed that for wide networks in NTP/SP trained with MSE loss and SGD, this critical learning rate is $2/\lambda_0^H$ early in training. Further investigation by Kalra and Barkeshli [18] demonstrated that sharpness reduction during early training causes $\eta_c$ to increase with depth and $1/\text{width}$. In such scenarios, $\eta_c$ can be as large as $40/\lambda_0^H$. Cohen et al. [5] demonstrated that sharpness at late training times for GD with momentum coefficient $\beta$ oscillates above $(2+2\beta)/\eta$, suggesting $\eta_c \gtrsim (2+2\beta)/\lambda_t^H$ at late training times. Expanding on this, Cohen et al. [6] analyzed adaptive optimizers and found that for Adam, the pre-conditioned sharpness $\lambda^{P^{-1}H}$ oscillates around $(2+2\beta_1)/\eta(1-\beta_1)$ at late training times. The instability threshold also depends on the mini-batch size [35] and is often observed to be smaller than their full batch counterparts [5, 6].

## B.3 Estimating the Instability Threshold

This section describes the method for estimating the instability threshold $\eta_c$ at initialization (or generically, any point $\theta_t$) using only forward passes. The method consists of two stages:

**Exponential Search:**   An exponential search, starting from an initial guess $\eta_0$, iteratively multiplies $\eta_0$ by a factor $k = 2$ until the loss increases. This identifies an interval $[\eta_{lwr}, \eta_{uppr}]$ containing $\eta_c$. The detailed algorithm is described in Algorithm 1. Unless specified, we use $\eta_0 = 10^{-4}$ as our initial guess. If the loss already increases at the initial guess $\eta_0$, we set $\eta_{init} = \eta_0$.

---

**Algorithm 1** Exponential Search

---

1: **Input:** (Initial weights: $\theta_0$, Initial guess: $\eta_0$)
2: **Output:** Interval $[\eta_{lwr}, \eta_{uppr}]$ containing $\eta_c$
3: Evaluate initial loss $L(\theta_0)$
4: Initialize $\eta \leftarrow \eta_0$
5: $\theta_1 \leftarrow \text{Optimizer}(\eta, \theta_0)$
6: Evaluate $L(\theta_1)$
7: **while** $L(\theta_1) < L(\theta_0)$ **do**
8:    **if** $\eta \geq \eta_{trgt}$ **then**
9:       $\eta \leftarrow \eta_{trgt}$
10:       **break**
11:    **end if**
12:    $\eta \leftarrow 2\eta$
13:    $\theta_1 \leftarrow \text{Optimizer}(\eta, \theta_0)$
14:    Evaluate $L(\theta_1)$
15: **end while**
16: $\eta_{uppr} \leftarrow \eta$
17: $\eta_{lwr} \leftarrow \eta/2$
18: **return** $[\eta_{lwr}, \eta_{uppr}]$

---

**Binary search:**   A binary search further narrows down $[\eta_{lwr}, \eta_{uppr}]$ by evaluating the loss at the midpoint $\eta_{mid} = (\eta_{lwr} + \eta_{uppr})/2$. If the loss increases, $\eta_{uppr}$ is updated to $\eta_{mid}$; otherwise, $\eta_{lwr}$ is set to $\eta_{mid}$. This process is repeated until the loss in the next step $L(\theta_1)$ satisfies the condition $L(\theta_1) < L(\theta_0)(1 + \delta)$, for some $\delta > 0$. The algorithm is detailed in Algorithm 2. In all our experiments, we set $\delta = 0.1$.

---

**Algorithm 2** Binary Search

---

1: **Input:** (Initial weights: $\theta_0$, Tolerance: $\delta$, Initial search interval $[\eta_{lwr}, \eta_{uppr}]$)
2: **Output:** Estimate of $\eta_c$
3: Evaluate $L(\theta_0)$
4: $\theta_1 \leftarrow \text{Optimizer}(\eta_{uppr}, \theta_0)$
5: Evaluate $L_{uppr} \leftarrow L(\theta_1)$
6: **while** $L_{uppr} > L(\theta_0)(1 + \delta)$ **do**
7:    $\eta_{mid} \leftarrow (\eta_{lwr} + \eta_{uppr})/2$
8:    $\theta_{mid} \leftarrow \text{Optimizer}(\eta_{mid}, \theta_0)$
9:    Evaluate $L_{mid} \leftarrow L(\theta_{mid})$
10:    **if** $L_{mid} < L(\theta_0)$ **then**
11:       $\eta_{lwr} \leftarrow \eta_{mid}$
12:    **else**
13:       $\eta_{uppr} \leftarrow \eta_{mid}$
14:       $L_{uppr} \leftarrow L(\theta_{mid})$
15:    **end if**
16: **end while**
17: **return** $\eta_c \leftarrow \eta_{uppr}$

---

While both $\eta_0$ and $\delta$ are additional hyperparameters, the method does not heavily depend on these choices. A poor initial guess of $\eta_0$ would only take a few more iterations to find an interval $[\eta_{lwr}, \eta_{uppr}]$.

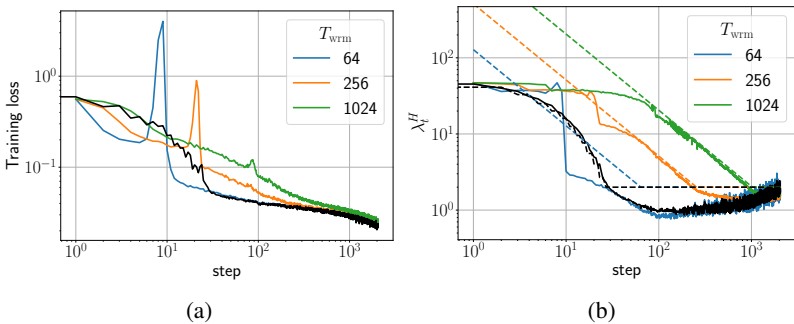

Figure 7: Comparison of persistent catapult warmup (in black) with linear warmup with different durations. The experimental setup is the same as in Figure 1, but the model is trained on the entire CIFAR-10 dataset using SGD with a batch size $B = 512$.

Meanwhile, any small value of $\delta \in (0, 1]$ is effective in finding $\eta_c$, as small initial loss spikes have minimal impact on the overall dynamics. Note that for Adam, a small $\delta (\sim 0.01)$ has to be selected to ensure that we do not observe large catapults.

## C   Persistent Catapult Warmup

Our analysis also motivates a potential parameter-free warmup strategy, which we refer to as *persistent catapult warmup*. The central idea behind this strategy is to repeatedly induce catapults aimed to progressively reduce sharpness (or pre-conditioned sharpness), thereby facilitating training at higher learning rates. Given a target learning rate $\eta_{\text{trgt}}$, the strategy consists of the following steps:

1. Start with a 'stable' reference point $\theta^*$, defined as a point where the loss decreases in the next step and estimate the interval $[\eta_{\text{lwr}}, \eta_{\text{uppr}}]$ containing $\eta_c$, as described in Appendix B.3.

2. Induce a catapult by increasing the learning rate to $\eta = \eta_{\text{uppr}}$.

3. Continue training and wait until the loss falls below the reference point, i.e., $L(\theta_t) < L(\theta^*)$. This new point now becomes the stable reference point.

4. Repeat the above steps until the target learning is achieved, i.e., $\eta = \eta_{\text{trgt}}$.

Here, the initial stable reference point is the model's initialization. The detailed algorithm is described in Algorithm 3 in Appendix C.

Figure 7 compares persistent catapult warmup (shown in black) with linear warmup. The persistent catapult warmup facilitates training at higher learning rates without the need to specify warmup duration. Since $\eta_c$ serves as an indicator of sharpness, persistent catapult warmup utilizes the local sharpness information to automatically determine the warmup rate, resulting in an adaptive non-linear warmup. This adaptive approach eliminates the need for manual tuning of the warmup duration, allowing for a more efficient and effective warmup.

Although persistent catapult warmup is a promising approach to warmup, it requires specifying how large a catapult should be induced, which introduces another hyperparameter. Nevertheless, persistent catapult warmup motivates the development of parameter-free warmup strategies that could simplify the training process. We leave further development of parameter-free warmup to future work.

## D   Experimental Details

This section provides additional experimental details. All models were implemented using the JAX [3], and Flax libraries [15]. The key results can be reproduced using the GitHub repo: `https://github.com/dayal-kalra/why-warmup`.

---

**Algorithm 3** Persistent Catapult Warmup

---

1: **Input:** (Initial weights: $\boldsymbol{\theta}_0$, Target learning rate: $\eta_{\text{trgt}}$, Tolerance: $\delta$)
2: $\theta^* \leftarrow \theta_0$ // Reference point
3: **while** $\eta < \eta_{\text{trgt}}$ **do**
4:     **if** $L(\theta_t) < L(\theta^*)$ **then**
5:         Estimate $[\eta_{\text{uppr}}, \eta_{\text{lwr}}]$ containing $\eta_c$ using Algorithms 1 and 2 with tolerance $\delta$
6:         $\eta \leftarrow \eta_{\text{uppr}}$
7:         $\theta^* \leftarrow \theta_t$
8:     **else**
9:         continue
10:     **end if**
11: **end while**

---

### D.1 Datasets Details

#### D.1.1 Image Classification Tasks

We consider standard image classification datasets such as CIFAR-10, CIFAR-100 [21], and Tiny-ImageNet [1]. The images are normalized to have zero mean and unit variance. For MSE loss, we use one-hot encoding for the labels.

**Data augmentation:** For various image classification tasks, we employ data augmentation techniques, applied in the following order: random horizontal flips, random cropping, and mixup [41].

#### D.1.2 Language Modeling Tasks

We consider the next token prediction task on the Wikitext-2 and Wikitext-103 datasets [25]. The Wikitext-2 dataset consists of $\sim 2M$ tokens, whereas the Wikitext-103 dataset has $\sim 0.1B$ tokens. We use Byte Pair Encoding (BPE) tokenizer [32] with a Whitespace pre-tokenizer. For Wikitext-103 experiments, we considered a standard vocabulary size of $50,257$, whereas our initial Wikitext-2 experiments employed a smaller vocabulary size of $4096$.

### D.2 Model Details

This section describes the models considered, including their parameterization and initialization details. We adopt parameterizations outlined in Table 9 of Ref. [38]. Unless otherwise specified, we employ ReLU non-linearities and initialize the weights with a truncated normal distribution [1], with a variance $\sigma_w^2 = 2.0$ in appropriate parameterizations (details below), except for the last layer, which has a weight variance of $\sigma_w^2 = 1.0$. All biases are initialized to zeros.

#### D.2.1 Parameterizations

**Standard Parameterization (SP):** For SP, the weights are initialized with truncated Gaussian distribution $\mathcal{N}(0, \sigma_w^2/\text{fan}_{\text{in}})$ and the biases are initialized to zero.

**Maximal Update Parameterization ($\mu$P):** For $\mu$P, different schemes are employed for the intermediate and last layers. The intermediate layers are initialized using $\mathcal{N}(0, \sigma_w^2/\text{fan}_{\text{out}})$ and the layer outputs are scaled by the factor $\sqrt{\text{fan}_{\text{out}}/\text{fan}_{\text{in}}}$. In comparison, the layer weights are initialized with $\mathcal{N}(0, \sigma_w^2/\text{fan}_{\text{in}})$, and the final output is rescaled by the factor $\sqrt{1/\text{fan}_{\text{in}}}$. Conveniently, for SGD, the learning rate does not scale with width in the above $\mu$P formulation. In comparison, for Adam, the learning rate corresponding to input, intermediate, and output layers are rescaled by the factors $1/\sqrt{\text{fan}_{\text{out}}}$, $1/\sqrt{\text{fan}_{\text{in}}}$ and $1/\text{fan}_{\text{in}}$. Since we are utilizing $\mu$P only to obtain flat initializations, we omit the additional scaling of the learning rate for Adam in some experiments (e.g., Figure 2). As a result, the instability threshold is only dependent on the target learning rate $\eta_{\text{trgt}}$ during late training, rather than on the largest learning rate across layers. We refer to this parameterization as 'simple-$\mu$P' for Adam.

---

[1]for details, see `https://jax.readthedocs.io/en/latest/_autosummary/jax.nn.initializers.truncated_normal.html`

### D.2.2 Architectures

**Fully Connected Networks (FCNs):** We consider fully connected networks with a constant width of $n$ and a depth of $d$ layers. These networks are denoted by FCN-$d$-$n$. Unless specified, we considered $d = 4$ layer FCNs with width $n = 512$.

**WideResNets (WRNs):** We consider WideResNets [39] with $d$ layers, $S$ stages, and a widening factor of $k$, denoted by WRN-$d$-$k$. The number of channels in each stage $s \in [0, S)$ is given by $2^s \times 16 \times k$, with the input layer having 16 channels. For example, WRN-16-4 consists of $S = 3$ stages, each with $[2, 2, 2]$ layers, and the corresponding number of channels in each stage is $[64, 128, 256]$. In all our experiments, we use LayerNorm instead of BatchNorm.

**Transformers:** We consider Transformers with GPT-2 style architecture [30]. These models use sinusoidal positional embeddings [34] and are implemented in the Standard Parameterization (SP) with GELU activation [16]. We initialize all layers using the $\sigma_w^2/\text{fan}_\text{in}$ scheme, except for the embedding layers, as they do not involve matrix multiplication [9]. We consider both Pre-LN [36] and Post-LN [34] Transformer variants. We denote a Transformer with $d$ blocks and an embedding dimension of $n$ as LM-$d$-$n$. Unless specified, the model has $d = 4$ blocks, embedding dimension $n = 128$, context length $T_\text{cntxt} = 64$ and are trained for $10^4$ steps.

### D.3 Optimization Details

### D.3.1 Optimizers

**SGD(-M):** Given gradients $g_t$ at step $t$, Stochastic Gradient Descent with momentum (SGD-M) updates the parameters $\theta_t$ using learning rate $\eta_t$ and momentum $m_t$ with coefficient $\beta$. The update equations are:

$$m_{t+1} = g_t + \beta m_t, \tag{4}$$
$$\theta_{t+1} = \theta_t - \eta_t m_{t+1}. \tag{5}$$

Here, $\beta = 0$ corresponds to SGD. In all experiments incorporating momentum, the default value of the coefficient is set to $\beta = 0.9$.

**Adam:** Given gradients $g_t$ at step $t$, Adam [20] updates the parameters $\theta_t$ using learning rate $\eta_t$ and the first two moments of the gradient $m_t$ and $v_t$ with their coefficients $\beta_1$ and $\beta_2$, respectively. The equations governing the updates are:

$$m_{t+1} = \beta_1 m_t + (1 - \beta_1)g_t, \tag{6}$$
$$v_{t+1} = \beta_2 v_t + (1 - \beta_2)g_t^2, \tag{7}$$
$$\theta_{t+1} = \theta_t - \eta_t \frac{\hat{m}_{t+1}}{\sqrt{\hat{v}_{t+1}} + \epsilon}, \tag{8}$$

where $\hat{m}_t = \frac{m_t}{1 - \beta_1^t}$ and $\hat{v}_t = \frac{v_t}{1 - \beta_2^t}$ are the bias-corrected moments, and $\epsilon$ is a small scalar used for numerical stability. The pre-conditioner for Adam is given by:

$$P_{t+1} = (1 - \beta_1^{t+1}) \left[ \text{diag} \left( \sqrt{\frac{v_{t+1}}{1 - \beta_2^{t+1}}} \right) + \epsilon \mathbf{I} \right]. \tag{9}$$

In all experiments, the default values are set to $\beta_1 = 0.9$, $\beta_2 = 0.999$, and $\epsilon = 10^{-8}$, unless otherwise specified.

**GI-Adam:** Given gradients $g_t$ at step $t$, GI-Adam updates the parameters like Adam but with the second moment initialized using the gradients at initialization, i.e., $v_0 = g_0^2$. The equations governing the updates are:

$$m_{t+1} = \beta_1 m_t + (1 - \beta_1) g_t, \tag{10}$$

$$v_{t+1} = \beta_2 v_t + (1 - \beta_2) g_t^2, \tag{11}$$

$$\theta_{t+1} = \theta_t - \eta_t \frac{m_{t+1}}{\sqrt{\hat{v}_{t+1}} + \epsilon}, \tag{12}$$

where $\hat{v}_t = \frac{v_t}{1 - \beta_2^t}$ is the bias-correction second moment and $\epsilon$ is a small scalar used for numerical stability.

**Flat-Adam:** Given gradients $g_t$ at step $t$, Flat-Adam updates the parameters $\theta_t$ using learning rate $\eta_t$ and the first two moments of the gradients $m_t$ and $v_t$ with their coefficients $\beta_1$ and $\beta_2$, respectively. In contrast to Adam, we do not apply bias correction for the first moment and initialize the second moment at initialization using the gradients $v_0 = g_0^2$. The equations governing the updates are:

$$m_{t+1} = \beta_1 m_t + (1 - \beta_1) g_t, \tag{13}$$

$$v_{t+1} = \beta_2 v_t + (1 - \beta_2) g_t^2, \tag{14}$$

$$\theta_{t+1} = \theta_t - \eta_t \frac{m_{t+1}}{\sqrt{\hat{v}_{t+1}} + \epsilon}, \tag{15}$$

where $\hat{v}_t = \frac{v_t}{1 - \beta_2^t}$ is the bias-correction second moment and $\epsilon$ is a small scalar used for numerical stability. The pre-conditioner for Flat-Adam is given by:

$$P_{t+1} = \left[ \text{diag}\left( \sqrt{\frac{v_{t+1}}{1 - \beta_2^{t+1}}} \right) + \epsilon \mathbf{I} \right]. \tag{16}$$

### D.3.2 Linear Warmup

Warmup linearly increases the learning rate from an initial value $\eta_{\text{init}}$ to a target value $\eta_{\text{trgt}}$ over $T_{\text{wrm}}$ training steps. The learning rate $\eta_t$ at step $t$ is given by:

$$\eta_t = \eta_{\text{init}} + (\eta_{\text{trgt}} - \eta_{\text{init}}) \left( \frac{t}{T_{\text{wrm}}} \right). \tag{17}$$

Here, $\alpha := \frac{(\eta_{\text{trgt}} - \eta_{\text{init}})}{T_{\text{wrm}}}$ is referred to as the rate of warmup. Under the above definition, constant learning rate training corresponds to $T_{\text{wrm}} = 1$. $T_{\text{wrm}} = 1$ corresponds to constant learning rate. Unless otherwise specified, we set $\eta_{\text{init}} = 0$ when referring to linear warmup.

### D.3.3 Learning Rate Decay

In several experiments, we employ learning rate decay following the warmup phase. Specifically, we use cosine learning rate decay, which is detailed below.

**Cosine Decay:** Towards the end of training, it is typical to reduce the learning rate to a small value. Cosine decay is a commonly used method for decaying the learning rate from an initial value of $\eta_{\text{trgt}}$ down to a value $\eta_{\text{min}}$ over $T_{\text{cos}}$ steps, according to the rule:

$$\eta_t = \eta_{\text{trgt}} + (\eta_{\text{min}} - \eta_{\text{trgt}}) \left[ \frac{1}{2} \left( 1 + \cos\left( \frac{\pi t}{T_{\text{cos}}} \right) \right) \right]^{\rho}, \tag{18}$$

where $\rho$ governs the rate of decay, with $\rho = 1$ being the standard. Note that with $\rho = 0$, the learning rate is not decayed and instead maintained at $\eta_{\text{trgt}}$. In the above expression, $t$ counts the steps from the initiation of cosine decay and not the current training step. As per standard practice, we consider $\rho = 1$ and decay the learning rate to $\eta_{\text{min}} = \eta_{\text{trgt}}/10$.

### D.3.4 Target Learning Rate Sampling for Phase Diagrams

For SGD, target learning rates $\eta_{\text{trgt}}$ are exponentially sampled using the initial sharpness $\lambda_0^H$. Starting with $\eta_{\text{trgt}} = 1/\lambda_0^H$, subsequent rates are sampled until divergence as $2^x/\lambda_0^H$ for values of $x$ increased in integer steps starting from zero. For WRNs trained with Adam, we sample target learning rates exponentially as $\eta_{\text{trgt}} = 2^x \times 10^{-5}$, where $x$ is incremented in integer steps starting from zero until training failure. For Transformers, we sample the learning rate in a similar fashion but starting from $10^{-4}$ and increment $x$ in steps of $0.5$.

### D.4 Sharpness and Pre-conditioned Sharpness Measurement

We measured sharpness / pre-conditioned sharpness using the JAX implementation of the LOBPCG sparse eigenvalue solver with the tolerance set to $10^{-9}$ and maximum number of iterations to $n_{\text{iter}} = 1000$. In most cases, the solver converges within $40$ iterations. We performed these computations in float64, as the solver would not converge with float32 in some cases.

In certain instances, the pre-conditioned sharpness computation did not converge within 1000 solver iterations. Moreover, we observed that the solver converges on restarting it with a new initial guess of the eigenvector within $40$ iterations. To address these edge cases, we employed the following method: if the solver did not converge within $100$ iterations, we restarted it with a new initial guess for the eigenvector. We allowed for at most $10$ restarts with the maximum number of iterations set to $n_{\text{iter}} = 1000$ in the last attempt. In all reported cases, the solver converges using this method.

### D.5 Additional Figure Details

**Figure 1:** Training trajectories of 4-layer FCNs with width $n = 512$, trained on a 5k subset of CIFAR-10 using MSE loss and GD in (top) $\mu$P with $\eta_{\text{trgt}} = 1/\lambda_0^H$, where $\lambda_0^H \approx 0.05$, and (bottom) SP with $\eta_{\text{trgt}} = 32/\lambda_0^H$, where $\lambda_0^H \approx 50$.

**Figure 2:** Training loss and sharpness trajectories of $4$ layer FCNs with width $n = 512$, in (top) $\mu$P with learning rate $\eta_{\text{trgt}} = 0.003$ and (bottom) SP with $\eta_{\text{trgt}} = 0.001$ trained the CIFAR-10 dataset with MSE loss using full batch Adam with $\beta_1 = 0.9$, $\beta_2 = 0.999$ and $\epsilon = 10^{-8}$. In these experiments, we use data augmentation as described in Appendix D.1.1.

**Figure 3:** Test accuracy heatmaps of WRN-16-4 trained on CIFAR-10 using different parameterizations and loss functions using SGD with a batch size $B = 128$: (a) SP and MSE loss, (b) $\mu$P and cross-entropy loss (c) SP and cross-entropy loss. All models are trained for $10^5$ steps. In these experiments, we use data augmentation as described in Appendix D.1.1.

**Figure 4:** Test loss heatmaps of Pre-LN Transformers in SP trained on WikiText-103 with cross-entropy loss using (a) Adam, and (b) Flat-Adam (introduced in Section 6) over Adam. The Transformer models have $d = 4$ blocks, embedding dimension $n = 768$, a context length of $T_{\text{cnxt}} = 64$, and a batch size $B = 64$. These experiments also employ cosine decay and weight decay with $\lambda = 10^{-4}$.

**Figure 5:** Heatmaps showing (a) $T_{\text{reach}}$, number of steps to reach $\eta_{\text{trgt}}$, and (b) $T_{\text{save}}$, the effective number of steps saved on setting $\eta_{\text{init}} = \eta_c$ for WRN-16-4 in SP trained on CIFAR-10 with cross-entropy loss using SGD with $B = 128$ for $10^4$ steps. For a fair comparison with linear warmup, we choose $\eta_0 = \eta_{\text{trgt}}/T_{\text{wrm}}$ as our initial guess.

**Figure 6:** Training loss and sharpness trajectories of FCNs in SP. The experimental setup is identical to Figure 2 but with GI-Adam instead of standard Adam.

### D.6 Estimation of Computational Resources

The phase diagram experiments typically required about an hour on per run on an A100 GPU. Consequently, each phase diagram consumed approximately 100 A100 hours of computational time. With a total of 16 phase diagrams, this equates to 1600 A100 hours dedicated solely to phase diagram computations. Additionally, the warmup mechanism experiments, which were conducted over 2000

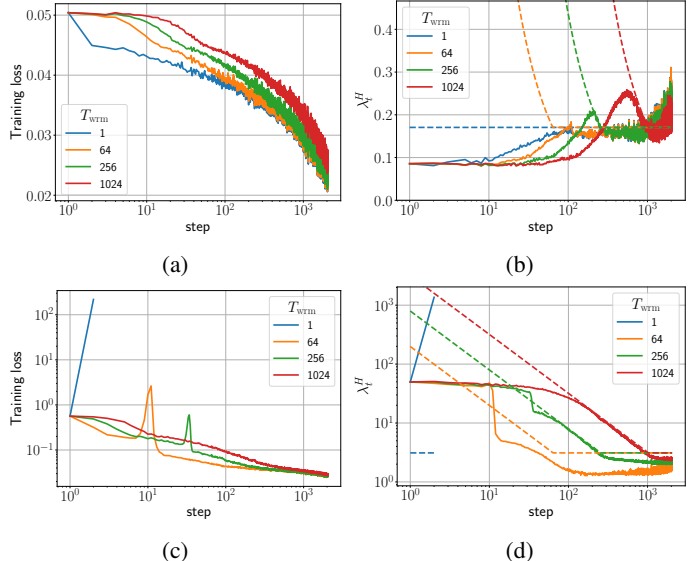

(a)

(b)

(c)

(d)

Figure 8: Training loss and sharpness trajectories of FCNs trained on CIFAR-10 with MSE loss using SGD with a batch size $B = 512$. The dashed lines in the sharpness figures illustrate the instability thresholds $^2/\eta_t$. (top) $\mu$P with learning rate $^1/\lambda_0^H$, (bottom) SP with learning rate $^{32}/\lambda_0^H$.

steps, required sharpness estimation. The FCN experiments required approximately 1200 A100 hours, while the WRN mechanism experiments consumed 1600 A100 hours. The experiments concerning the initial learning rate took about 20 A100 hours. This brings the total computational time amounted to approximately 4500 A100 hours. Preliminary experiments took about 1000 A100 hours. Hence, we estimate the total computational cost to be around 5500 A100 hours.

# E Additional Results for Mechanisms of Warmup

This section presents additional trajectories for warmup mechanisms discussed in Section 4 covering various architectures, loss functions, and optimizers.

## E.1 Stochastic Gradient Descent

Figure 8 shows that the warmup mechanisms for full batch GD are also observed in the SGD with a batch size $B = 512$. The results for other optimizers in the mini-batch setting are discussed in their respective sections.

## E.2 Stochastic Gradient Descent with Momentum

While the warmup mechanisms of SGD with momentum are fundamentally similar to those of vanilla SGD, three key differences arise, as discussed below.

First, the training loss can decrease loss in an oscillatory fashion during training [11]. To illustrate this, consider the full-batch GD with momentum. The middle row of Figure 9 demonstrates the sharpness reduction case with the learning rates well below the stability threshold ($\eta << \eta_c$). Despite being far below these thresholds, the loss does not decrease monotonically but converges in an oscillatory fashion. This makes it challenging to differentiate between warmup-induced catapults and fluctuations in loss due to the intrinsic effects of momentum. Nevertheless, we can still observe loss spikes correlated with an abrupt decrease in sharpness at large learning rates, as seen in the bottom row of the same figure. Similar to the SGD case, we observe these catapults are delayed and become smaller in magnitude on increasing the warmup duration.

Next, the stability threshold $\eta_c$ for SGD with momentum evolves during training. For simplicity of explanations, we again consider the full batch GD case. The stability threshold $\eta_c$ for SGD with

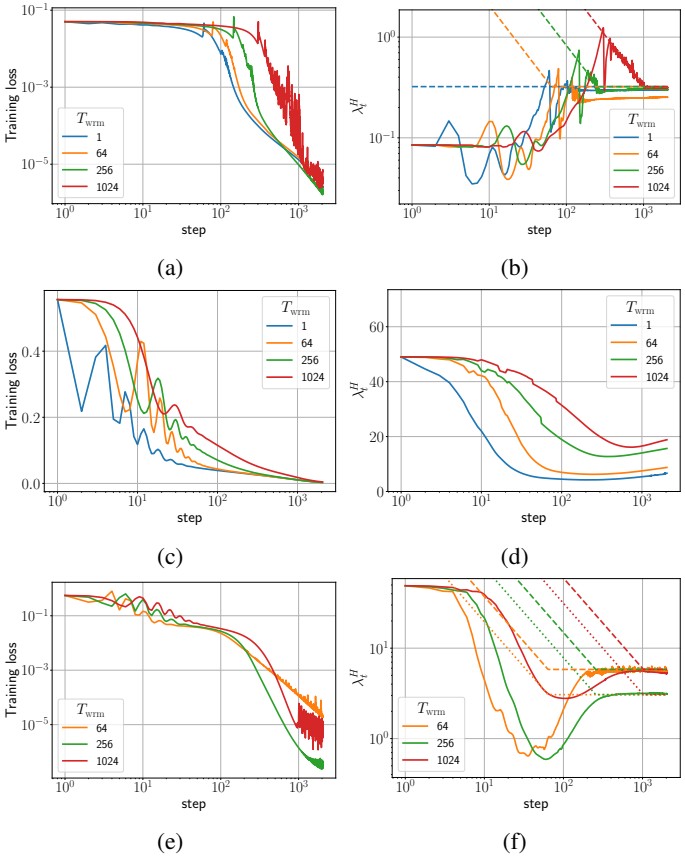

Figure 9: Training loss and sharpness trajectories of FCNs trained on 5k subset of CIFAR-10 using MSE loss and full batch GD with momentum $\beta = 0.9$: (top) $\mu$P with learning rate $1/\lambda_0^H$ (middle) SP with learning rate $1/\lambda_0^H$, and (bottom) SP with learning rate $32/\lambda_0^H$. The dotted lines in the sharpness figures correspond to the $(2+2\beta)/\eta_t$ curves, while dashed lines show the $2/\eta_t$ for reference.

momentum changes from $2/\lambda_0^H$ at initialization to $(2+2\beta)/\lambda_t$ late in training. At initialization, the momentum vector is set to zero $\boldsymbol{m}_0 = 0$, and the stability is given by vanilla GD threshold $2/\lambda_0^H$. As training progresses, the momentum $\boldsymbol{m}_t$ increases in magnitude, and the instability threshold at late training time becomes $(2+2\beta)/\lambda_t$. The bottom row of Figure 9 show the sharpness trajectories with both $2/\eta_t$ and $(2+2\beta)/\eta_t$ curves. For $T_{\text{wrm}} = 64$, the learning rate curve $2/\eta_t$ causes an abrupt decrease in sharpness, which is coupled with a loss spike. For longer warmup durations, the sharpness decreases before training exceeds the $2/\lambda_t^H$.

Finally, the instability threshold $\eta_c$ for SGD with momentum significantly decreases for smaller batch sizes. Figure 10 shows the training trajectories under the same setup as in Figure 8, but with momentum coefficient $\beta = 0.9$. The late-time sharpness trajectories oscillate well below the $(2+2\beta)/\eta_t$ (and even below $2/\lambda_t^H$), whereas the vanilla SGD counterpart oscillates on the $2/\eta_t$ curve. This indicates a strong dependence of the instability threshold on batch size.

Besides these three differences, we note that the warmup mechanisms of SGD with momentum are similar to the vanilla SGD case. We leave a thorough analysis of the early sharpness dynamics of SGD with momentum for future works.

### E.3 Stochastic Gradient Descent and Cross-entropy Loss

The warmup mechanisms for models trained with cross-entropy loss exhibit trends similar to those observed with MSE loss with one crucial difference. Near convergence, sharpness first increases and then abruptly decreases. The decrease in sharpness towards the end of training is observed in previous studies analyzing SGD with fixed learning rate [5]. Additionally, we observe higher

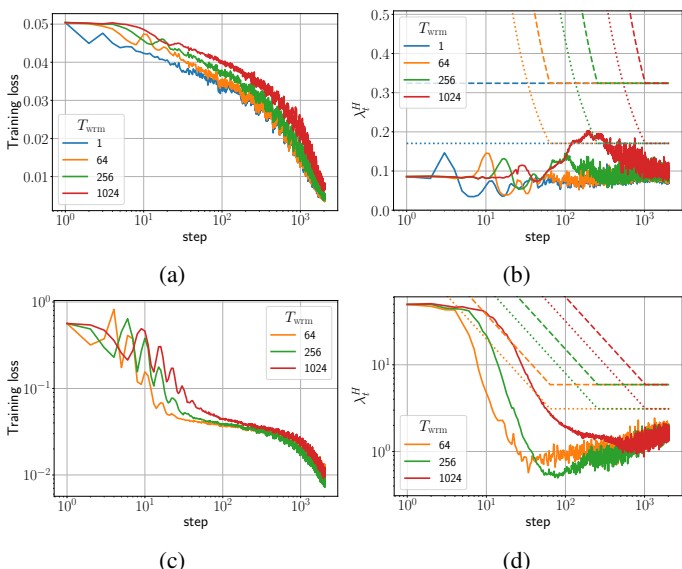

Figure 10: Training loss and sharpness trajectories of FCNs trained on CIFAR-10 with MSE loss using SGD with a batch size $B = 512$ and momentum $\beta = 0.9$: (top) $\mu$P with learning rate $1/\lambda_0^H$, and (bottom) SP with learning rate $32/\lambda_0^H$. The dotted lines in the sharpness figures correspond to the $(2+2\beta)/\eta_t$ curves, while dashed lines show the $2/\eta_t$ for reference. Similar mechanisms are observed for cross-entropy loss with a decrease in sharpness at late training times, as detailed in Appendix E.3.

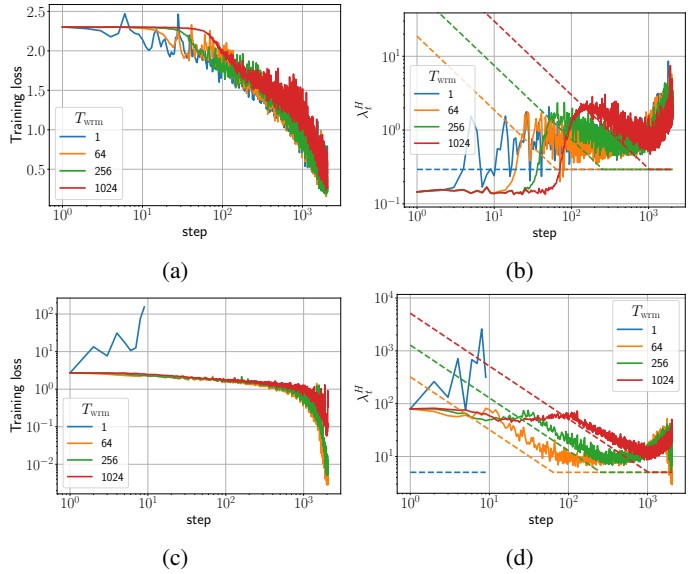

Figure 11: Training loss and sharpness trajectories of FCNs trained on CIFAR-10 with cross-entropy loss using SGD with a batch size $B = 512$. (Top row) $\mu$P with learning rate $1/\lambda_0^H$ (Bottom row) SP with learning rate $32/\lambda_0^H$.

fluctuations compared to the MSE loss case. Figure 11 shows trajectories of FCNs under different parameterizations trained on CIFAR-10 with cross-entropy loss using vanilla SGD. Meanwhile, Figure 12 shows the loss and sharpness trajectories of FCNs in SP trained on CIFAR-10 with cross-entropy loss using full batch GD with and without momentum.

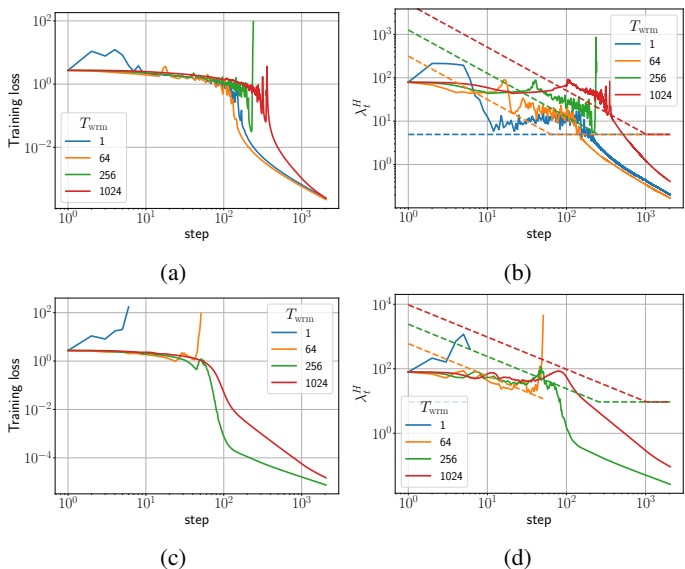

Figure 12: Training loss and sharpness trajectories of FCN-4-512 in SP trained on 5k subset of CIFAR-10 with cross-entropy loss using full batch GD with learning rate $^{32}/\lambda_0^H$ with momentum coefficient (top) $\beta = 0.0$ and (bottom) $\beta = 0.9$.

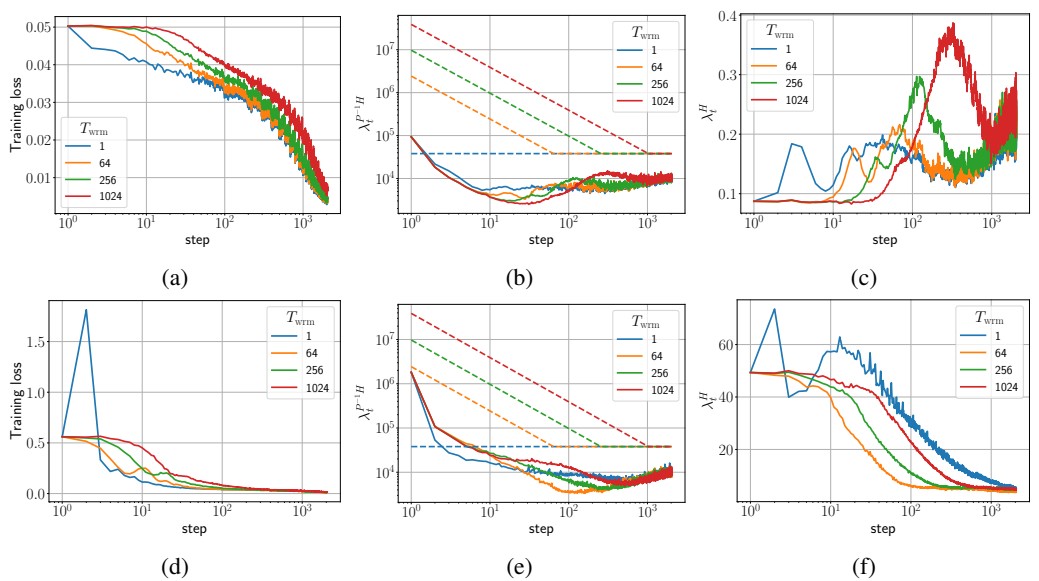

Figure 13: Training loss and sharpness trajectories of FCN-4-512 in (top) $\mu$P and (bottom) SP trained on CIFAR-10 with MSE loss using Adam with learning rate $\eta = 0.001$, batch size $B = 512$, $\beta_1 = 0.9$ and $\beta_2 = 0.999$. The dashed lines in the sharpness figures illustrate the instability thresholds $^{(2+2\beta_1)}/_{\eta_t(1-\beta_1)}$.

### E.4 Warmup Mechanisms of Adam

As discussed in Section 4.3, the instability threshold for Adam is determined by the pre-conditioned sharpness $\lambda^{P^{-1}H}$ and not by the sharpness itself. Moreover, training dynamics falls under the sharpness reduction case as the pre-conditioned sharpness starts off large and reduces considerably during the first few training.

Figure 13 shows the training trajectories of FCNs trained with Adam in the same setting as in Figure 2 but with a batch size of $B = 512$. Similar to the SGD with momentum case, the late

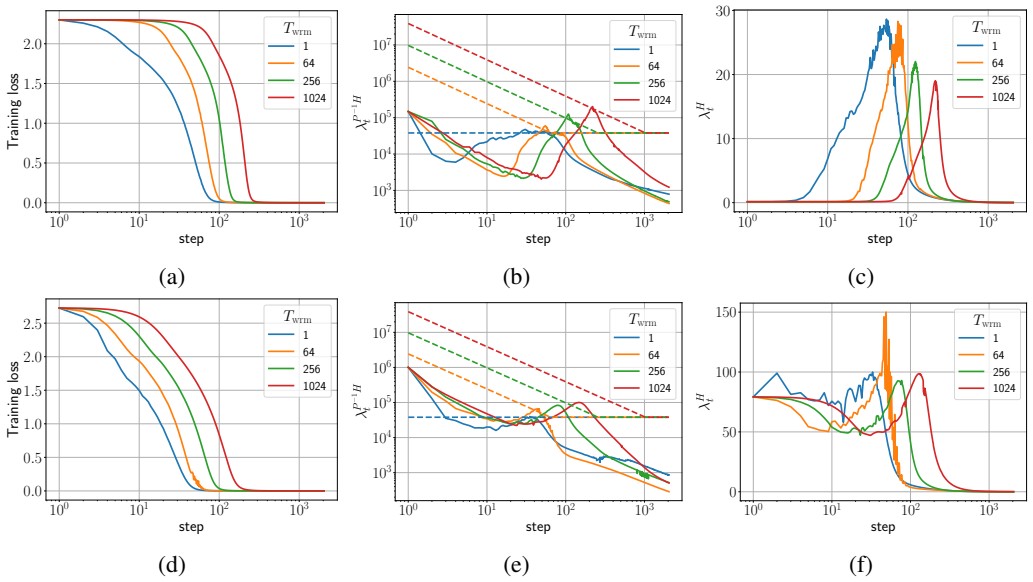

(a)        (b)        (c)

(d)        (e)        (f)

Figure 14: Training loss and sharpness trajectories of FCNs in (top) $\mu$P and (bottom) SP trained on CIFAR-10 with cross-entropy loss using full-batch Adam with learning rate $\eta = 0.001$, $\beta_1 = 0.9$ and $\beta_2 = 0.999$. The dashed lines in the sharpness figures illustrate the instability thresholds ${(2+2\beta_1)}/{\eta_t(1-\beta_1)}$.

time sharpness oscillates far below the instability threshold (${(2+2\beta_1)}/{\eta_t(1-\beta_1)}$), suggesting that the instability threshold heavily decreases with a smaller batch size. We note similar findings by Ref. [6].

Next, Figure 14 show the warmup mechanism of FCNs trained with cross-entropy loss using Adam under the full-batch setting. Similar to the SGD case, the pre-conditioned sharpness decreases towards the end of training.

## E.5 Different Architectures and Datasets

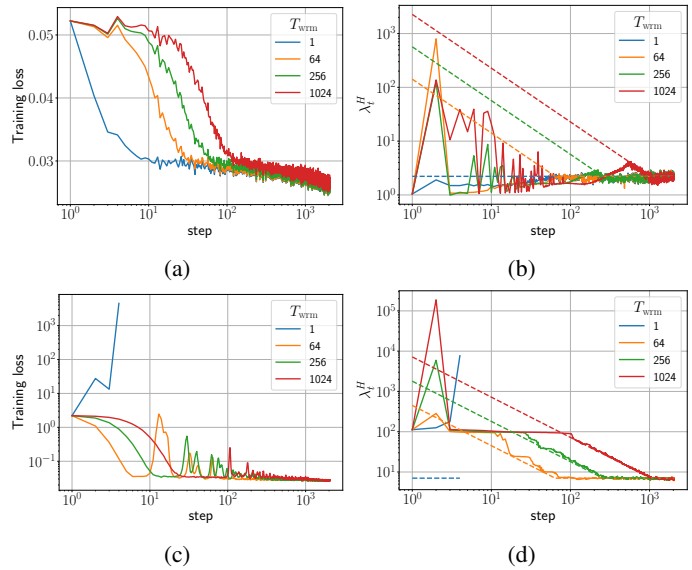

(a)        (b)

(c)        (d)

Figure 15: WRN-16-1 trained on CIFAR-10 with MSE loss using vanilla SGD with batch size $B = 512$: (top) $\mu$P with $\eta_{\text{trgt}} = {1}/{\lambda_0^H}$ and (bottom) SP with $\eta_{\text{trgt}} = {32}/{\lambda_0^H}$.

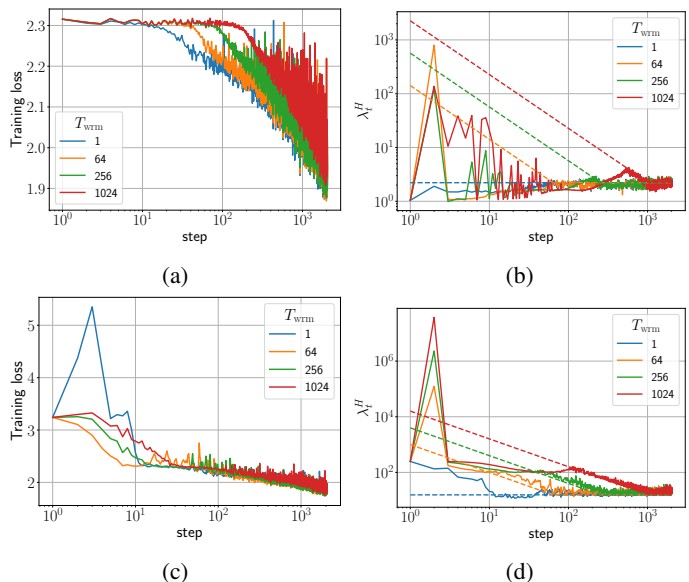

(a)         (b)

(c)         (d)

Figure 16: WRN-16-1 trained on CIFAR-10 with cross-entropy loss using vanilla SGD with batch size $B = 512$: (top) $\mu$P with $\eta_{\text{trgt}} = 1/\lambda_0^H$ and (bottom) SP with $\eta_{\text{trgt}} = 32/\lambda_0^H$.

In the previous sections, we confined our analysis to FCNs to thoroughly explore the effects of different optimizers and loss functions. This section expands on those results by demonstrating that the observed warmup mechanisms apply to ResNets and Transformers as well. The Resnet experiments also employ data augmentation as detailed in Appendix D.1.

Figures 15 and 16 show the training trajectories of WideResNets (WRNs) trained on CIFAR-10 with MSE and cross-entropy loss using SGD. These trajectories generally reflect the warmup mechanisms discussed in Section 4. However, certain additional features obscure the clarity of these mechanisms. Notably, we observed a significant sharpness spike on the first training step when using longer warmup durations, which automatically resolves in the subsequent step. The magnitude of this spike increases with longer warmup periods. Further analysis revealed that this phenomenon is associated with an initial increase in the first LayerNorm parameters, which also resolves automatically by the second step. Beyond this observation, the training trajectories align with the warmup mechanisms described in the main text.

Figure 17 illustrates the warmup mechanisms of Pre-LN Transformers trained on the WikiText-2 with SGD. The Pre-LN Transformer (top row) starts in a flat landscape region ($\lambda_0^H \sim 5$) and experiences progressive sharpening right from initialization. In contrast, when the last LayerNorm (just before the final linear layer) is removed (bottom row), the model starts training in a significantly sharper region, with the initial sharpness 100 times larger than the standard Pre-LN Transformer. This modified Pre-LN Transformer experiences a reduction in sharpness during the early stages of training.

Figure 18 presents the warmup mechanisms of Pre-LN Transformers trained on WikiText-2 using the Adam optimizer. Consistent with the results in the main text, the pre-conditioned sharpness exhibits a reduction early in training, despite the model initializing in a very flat region.

These experiments demonstrate that Transformers trained on language modeling tasks exhibit warmup mechanisms consistent with those discussed in the main text.

# F   Additional Phase Diagrams

This section presents further results related to the phase diagrams of warmup shown in Section 5.

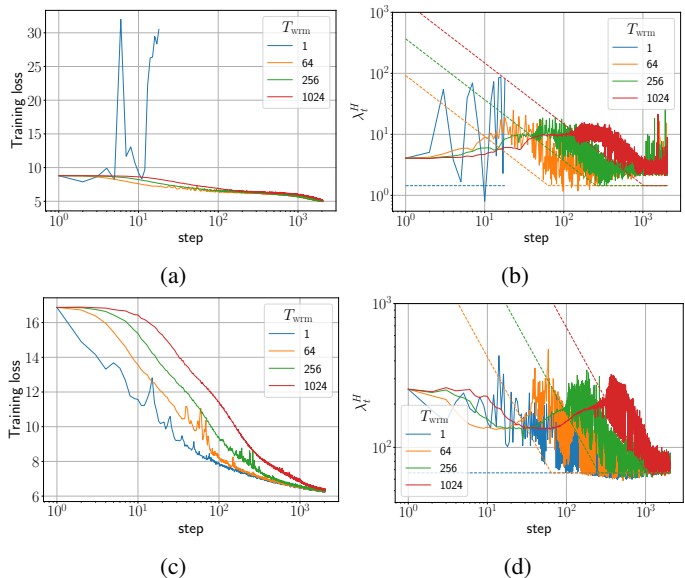

Figure 17: LM-4-128 trained on the WikiText-2 dataset with cross-entropy loss using SGD with a batch size $B = 512$ and a context length $T_{cntx} = 64$. The top row shows the warmup mechanisms of a Pre-LN Transformer with $\eta_{trgt} = {}^{5.65}/\lambda_0^H$, while the bottom row shows the results for the same Pre-LN Transformer but with the last LayerNorm removed and a learning rate of $\eta_{trgt} = {}^8/\lambda_0^H$.

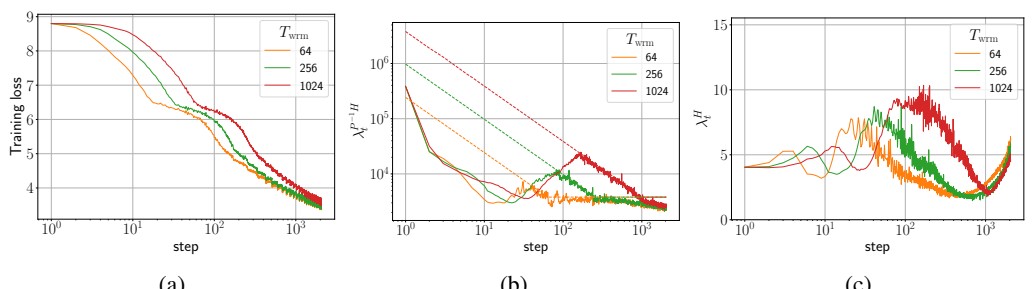

Figure 18: Pre-LN LM-4-128 trained on the WikiText-2 dataset with cross-entropy loss using Adam with a target learning rate $\eta_{trgt} = 0.003$, a batch size $B = 512$ and a context length $T_{cntx} = 64$.

## F.1 Phase Diagrams for different Models and Datasets

Figure 19 shows the test accuracy heatmaps of WRN-16-4 trained on CIFAR-100 and Tiny-ImageNet. These models are trained using cross-entropy loss using SGD with a batch size of $B = 128$. Additional phase diagrams for Adam are presented in Appendix F.3.

Figure 20(a) shows the test loss heatmaps of Pre-LN Transformer trained on the WikiText-2 dataset using SGD with a batch size $B = 64$. Figure 20(b) shows the Pre-LN Transformer under the same setup except for the last layer LayerNorm removed. The standard Pre-LN Transformer starts off with a small sharpness, while the version without the last LN starts off with 100 times higher curvature and requires warmup to achieve good performance.

## F.2 The Effect of Momentum and Learning Rate Decay

Figure 21 shows that incorporating momentum and cosine decay (for details, see Appendix D.3.3) minimally affects the warmup phase diagrams. While the conclusions regarding warmup presented in the main text remain unaffected, we note a few interesting observations.

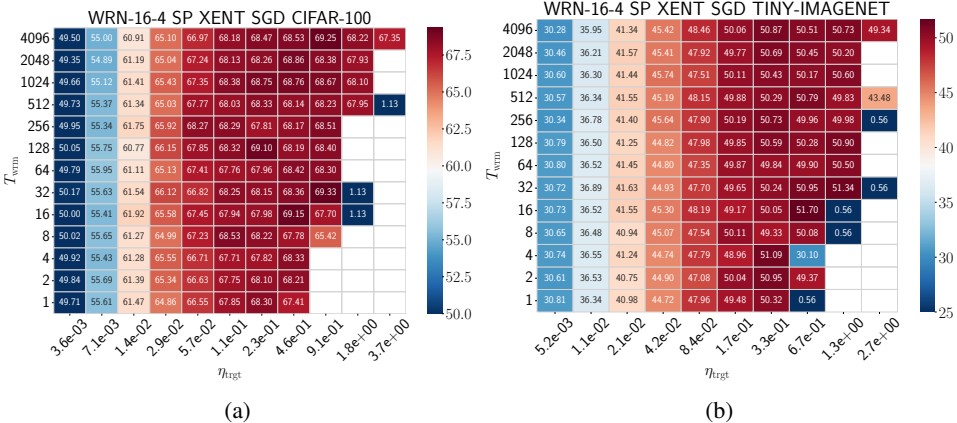

(a)                                                      (b)

Figure 19: Test accuracy heatmaps of WideResNets (WRNs) in SP trained on (a) CIFAR-100 and (b) Tiny ImageNet with cross-entropy loss using SGD with batch size $B = 128$.

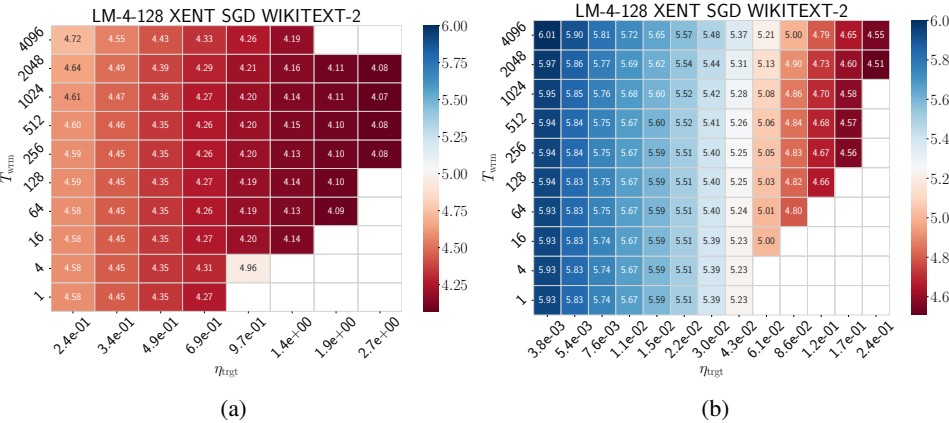

(a)                                                      (b)

Figure 20: Test loss heatmaps of LM-$4$-$128$ in SP trained on WikiText-2 with cross-entropy loss using SGD with a batch size $B = 64$: (a) Pre-LN Transformer and (b) Pre-LN Transformer without the last LayerNorm.

First, the divergent boundary shifts leftward on incorporating momentum, indicating that momentum permits smaller target learning rates without warmup, and warmup helps SGD-M more. Meanwhile, cosine decay has a minimal effect on the divergent boundary.

Additionally, we observe a performance enhancement by incorporating momentum, especially at small learning rates. In contrast, a decaying learning rate beyond warmup degrades performance at small learning rates while improving at higher ones. Finally, incorporating both momentum and cosine decay leads to further enhancement, indicating a synergistic interaction between the two.

### F.3 Phase Diagrams of Adam and GI-Adam

Figures 23 to 25 compare the warmup phase diagrams of Adam and GI-Adam of WRNs trained on CIFAR-100, Tiny-ImageNet and of Transformers trained on WikiText-2 and Wikitext-103 dataset. Similar to the results shown in the main text, GI-Adam enhances performance over standard Adam by pushing the failure boundary.

### F.4 Comparison with RAdam

Figures 26 and 27 and table 1 compare the performance of GI-Adam with RAdam. We observe that GI-Adam performs on par or better than RAdam while offering a simpler modification to Adam.

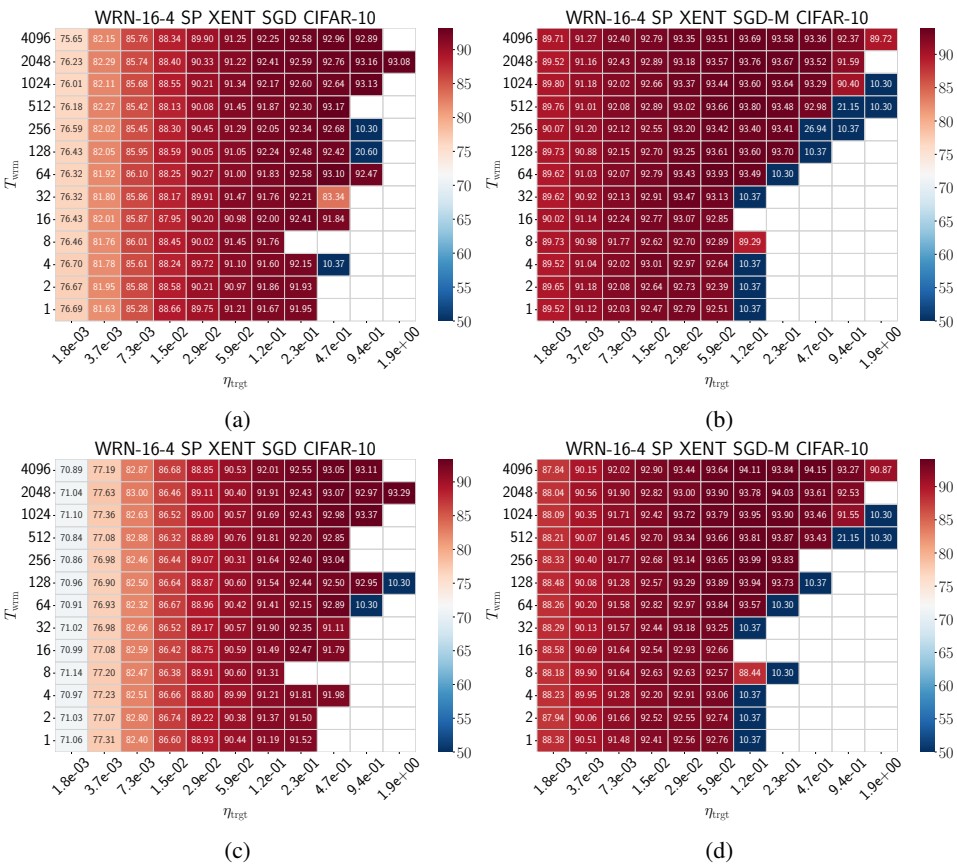

Figure 21: Test accuracy heatmaps of WideResNets (WRNs) in SP trained on CIFAR-10 with cross-entropy loss using SGD with batch size $B = 128$: (top row) no cosine decay (a) no momentum, (b) momentum with $\beta = 0.9$, and (bottom row) with cosine decay (c) no momentum, and (d) momentum with $\beta = 0.9$. The setting of (a) is the same as in Figure 3(c) but with a different mini-batch sequence.

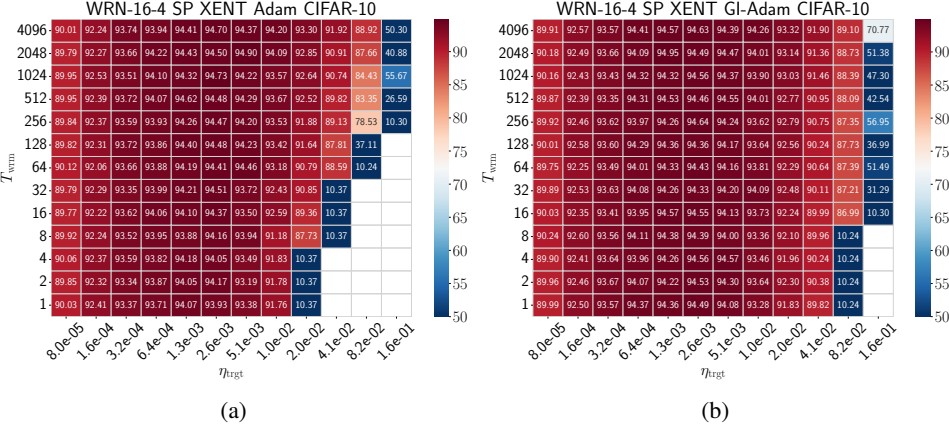

Figure 22: Test accuracy heatmaps of WRN-16-4 trained on CIFAR-100 with cross-entropy loss using (left) standard Adam, and (right) GI-Adam with batch size $B = 128$.

# G Non-divergence of Adam

Figure 28 shows that, despite experiencing catastrophic instabilities during early training, Adam does not diverge well beyond the training failure boundary. While Adam can recover from these

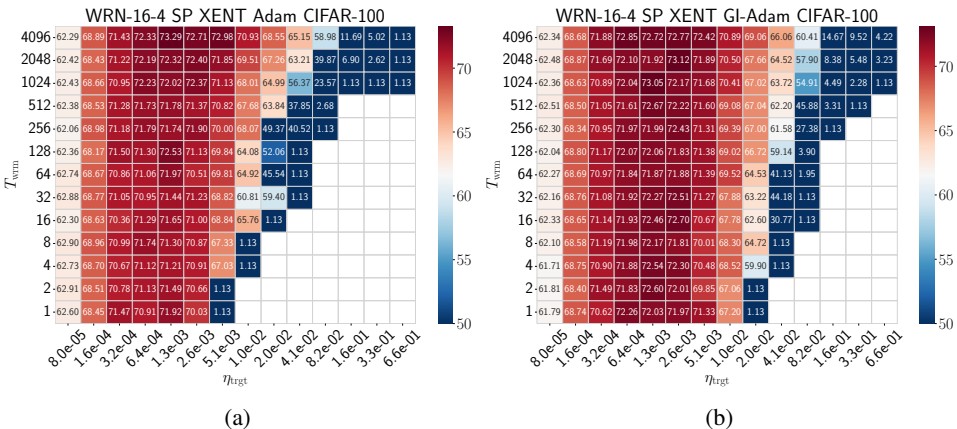

(a)  (b)

Figure 23: Test accuracy heatmaps of WRN-16-4 trained on CIFAR-100 with cross-entropy loss using (left) standard Adam, and (right) GI-Adam with batch size $B = 128$.

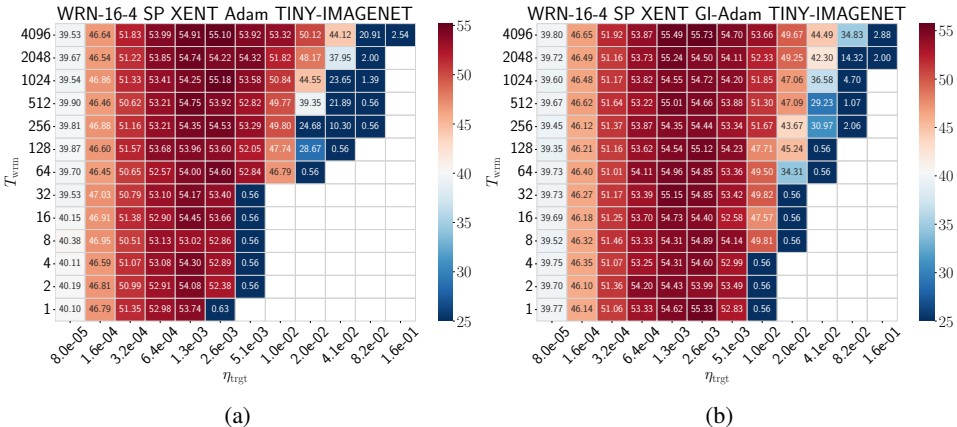

(a)  (b)

Figure 24: Test accuracy heatmaps of WRN-16-4 trained on Tiny-ImageNet with cross-entropy loss using (left) standard Adam, and (right) GI-Adam with batch size $B = 128$.

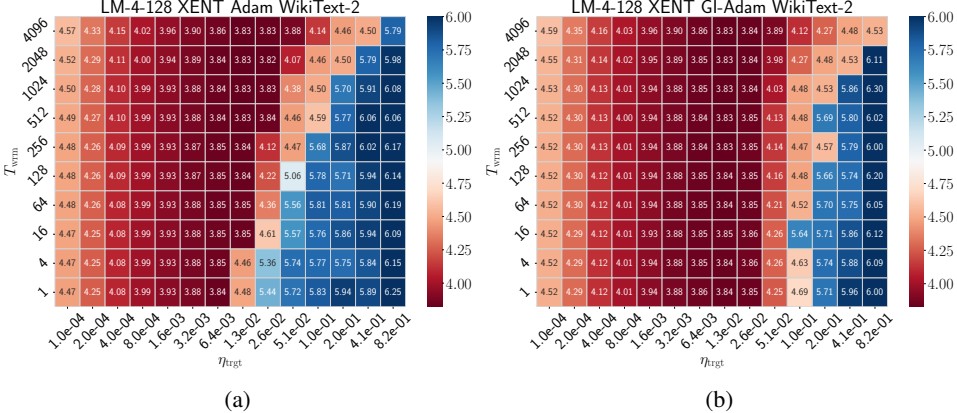

(a)  (b)

Figure 25: Test loss heatmaps of LM-4-128 in SP trained on WikiText-2 with cross-entropy loss using (a) standard Adam, and (right) GI-Adam with batch size $B = 64$.

instabilities, the model's performance is severely impacted, resulting in training failures rather than convergence to a reasonable minimum.

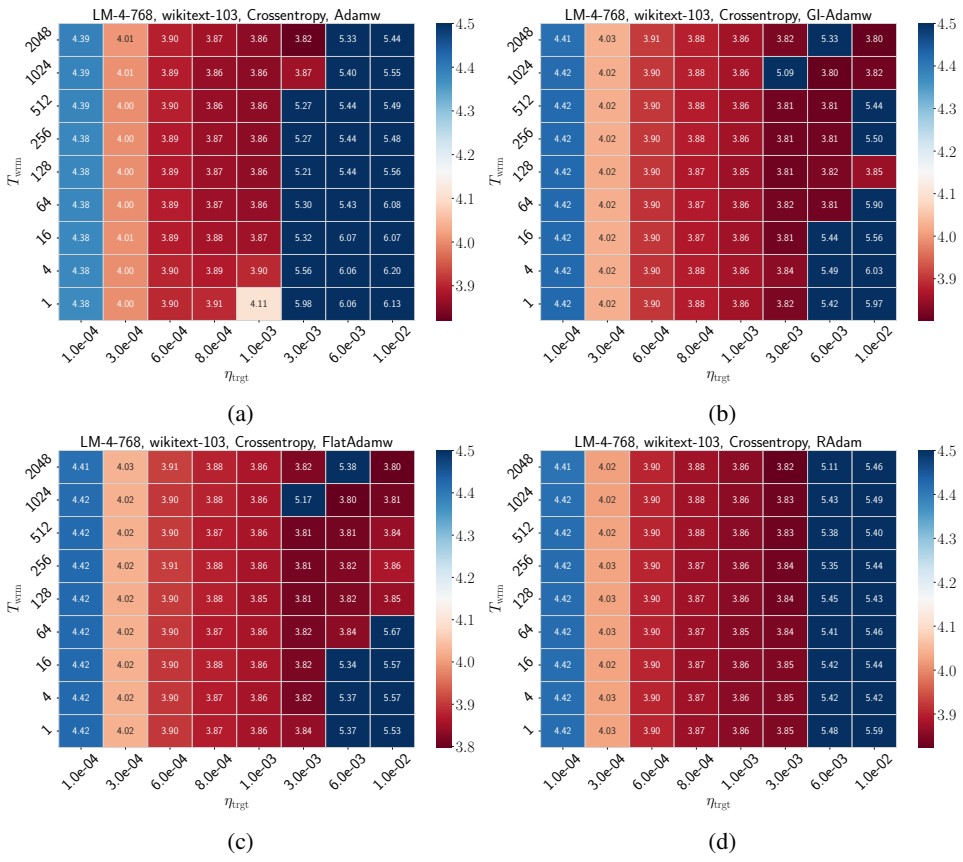

Figure 26: Test loss heatmaps of LM-4-768 in SP trained on WikiText-103 with cross-entropy loss using (a) standard Adam, (b) GI-Adam, (c) Flat-Adam, and (d) RAdam with batch size $B = 64$.

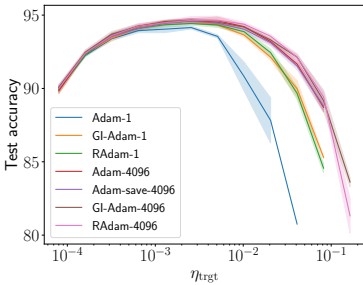

Figure 27: Comparison of optimizer performance on WRNs trained on CIFAR-10. The value next to the optimizer name corresponds to the warmup duration. Each value corresponds to an average over five initializations. Here, Adam-save corresponds to Adam with warmup $\eta_t = \eta_{\text{init}} + \eta_{\text{trgt}} \frac{t}{T_{\text{wrm}}}$.

These large loss catapults cause the gradients $g$ to spike during early training, leading to a substantial increase in its second moment $v$. While the gradients return to a lower value after a few training steps, the second moment remains large in magnitude for a prolonged period. These large values of $v$ result in a small effective learning rate, which hinders training to escape these high-loss regions. Consequently, the models remain stuck in a suboptimal state rather than converging. We refer to this as a training failure.

Upon closer examination of the individual layers during training failures, we found that certain layers or residual blocks output zero. This results in vanishing gradients except for the last layer bias and training halts. We defer the detailed analysis of Adam's failures to future work.

| Optimizer | Test Accuracy (mean ± std) |
|---|---|
| Adam-1 | $94.1515 \pm 0.0980$ |
| GI-Adam-1 | $94.6123 \pm 0.1273$ |
| Radam-1 | $94.4244 \pm 0.1052$ |
| Adam-4096 | $94.6163 \pm 0.2528$ |
| GI-Adam-4096 | $94.6499 \pm 0.1112$ |
| Adam-save-4096 | $94.6084 \pm 0.1664$ |
| Radam-4096 | $94.6460 \pm 0.2532$ |

Table 1: Performance comparison of different optimizers with varying warmup durations.

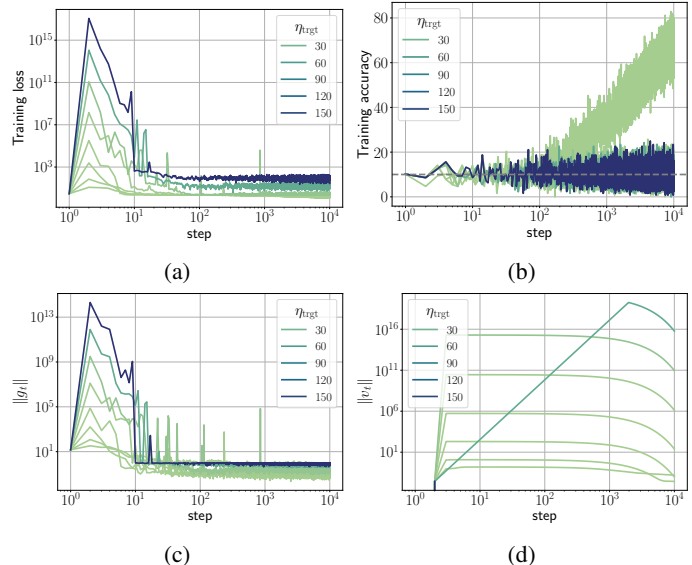

Figure 28: Training trajectories of WRNs trained on CIFAR-10 using Adam with cross-entropy loss and varying learning rates. The setup is identical to the $T_{\text{wrm}} = 1$ row of Figure 4, but without employing cosine learning rate decay. The first training failure is observed at a learning rate of $\eta_{\text{trgt}} = 0.02048$. To investigate the behavior beyond the training failure boundary, learning rates are sampled from $\eta_{\text{trgt}} = 0.01024$ (just below the failure boundary) up to $\eta_{\text{trgt}} \approx 150$.

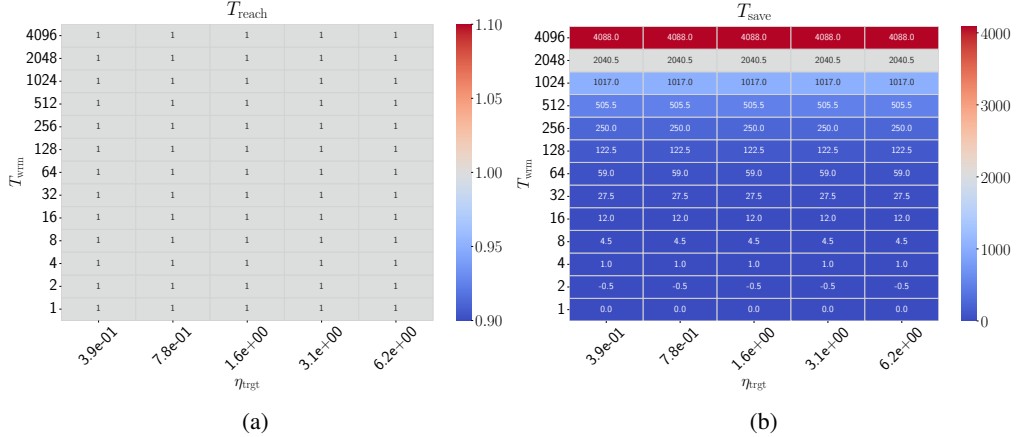

Figure 29: Heatmaps illustrating the computational savings in reaching $\eta_{\text{trgt}}$ for WRN-16-4 in $\mu$P trained on CIFAR-10, using SGD and cross-entropy loss. Colored cells show the warmup steps $T_{\text{reach}}$ (left) and the total steps saved $T_{\text{save}}$ (right), while the empty cells correspond to training divergence.

# H  Additional results for the Initial Learning Rate Selection

This section provides additional results for the initial learning rate selection. Figure 29 shows the number of steps $T_{\text{reach}}$ required to reach the target learning rate and the effective number of steps saved $T_{\text{save}}$ for WRNs in $\mu$P. Note that in such cases, $\eta_c \approx \eta_{\max}$. We observe that $T_{\text{reach}} = 1$ for a wide range of learning rates, saving almost the entire warmup duration. Figures 30 and 31 show similar phase diagrams for WRNs in SP and $\mu$P trained with Adam.

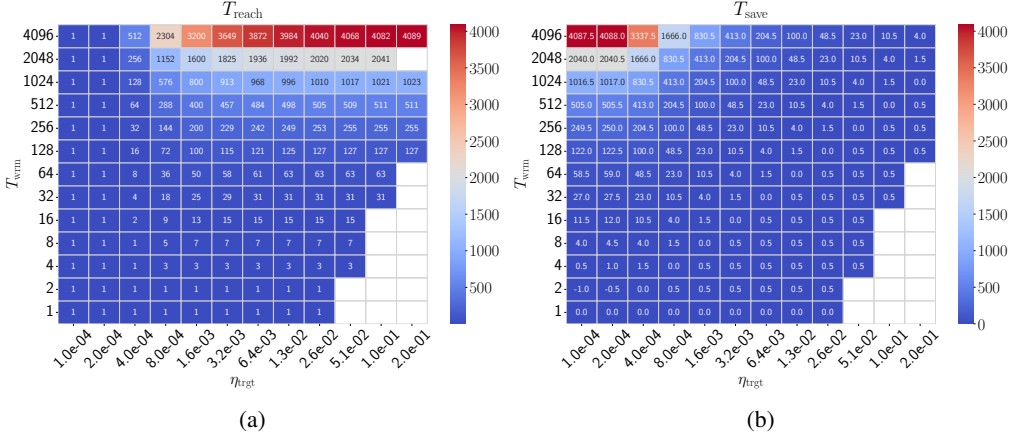

(a)                    (b)

Figure 30: Heatmaps illustrating the computational savings in reaching $\eta_{\text{trgt}}$ for WRN-16-4 in SP trained on CIFAR-10, using Adam and cross-entropy loss. Colored cells show the warmup steps $T_{\text{reach}}$ (left) and the total steps saved $T_{\text{save}}$ (right), while the empty cells correspond to training divergence.

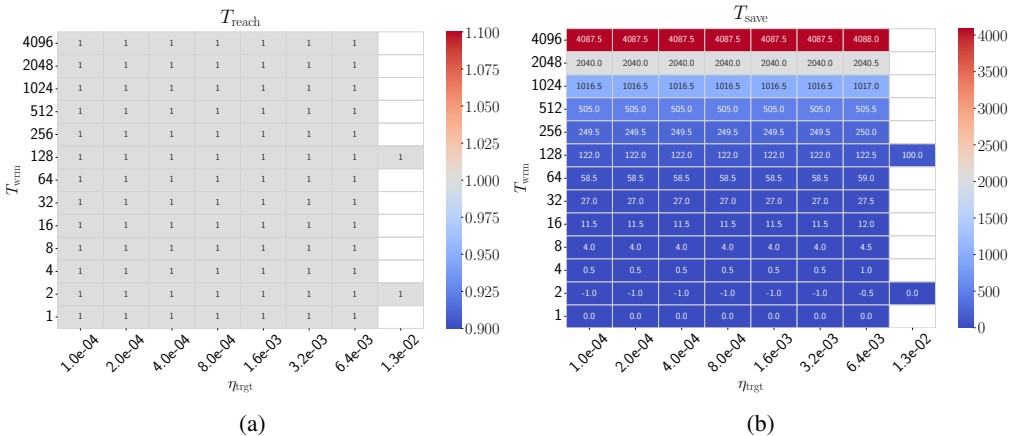

(a)                    (b)

Figure 31: Heatmaps illustrating the computational savings in reaching $\eta_{\text{trgt}}$ for WRN-16-4 in $\mu$P trained on CIFAR-10, using Adam and cross-entropy loss. Colored cells show the warmup steps $T_{\text{reach}}$ (left) and the total steps saved $T_{\text{save}}$ (right), while the empty cells correspond to training divergence.

# I  Additional Results on GI-Adam

This section presents additional results for GI-Adam. We provide further insights into the mechanisms and interpretations of GI-Adam.

## I.1  Warmup Mechanisms of GI-Adam

Figure 32 shows the training trajectories of FCNs with different parameterizations trained with GI-Adam. Notably, the pre-conditioned sharpness starts at significantly lower values than standard Adam. Specifically, for the $\mu$P model, the initial pre-conditioned sharpness $\lambda^{P^{-1}H}$ is around 2000

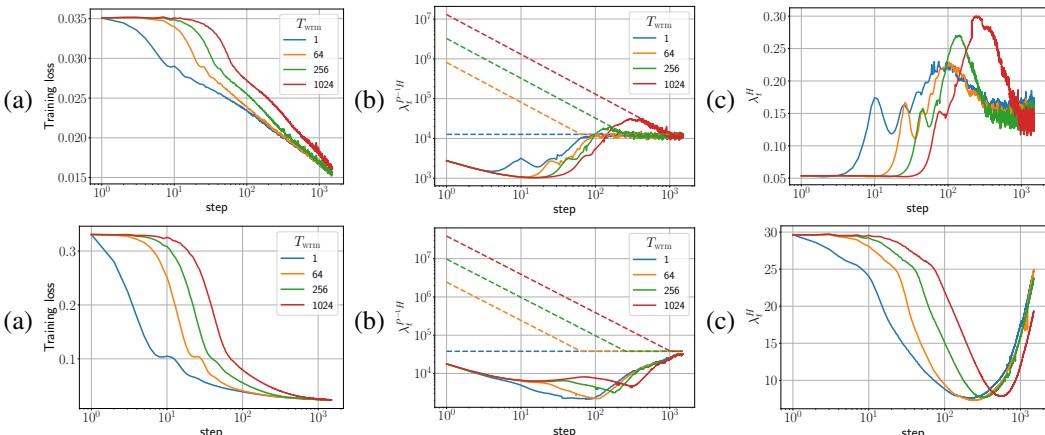

Figure 32: Training loss and sharpness trajectories of FCNs in (top) $\mu$P and (bottom) SP. The experimental setup is identical to Figure 2 but with GI-Adam instead of standard Adam.

instead of the value $10^5$ observed for Adam (c.f. Figure 2). Remarkably, this almost eliminates initial sharpness reduction. Similarly, the pre-conditioned sharpness for the SP model starts around $10^4$ instead of $10^6$. Notably, in the SP scenario, there is no initial spike in the $T_{\mathrm{wrm}} = 1$ (c.f. Figure 2), demonstrating that this simple modification effectively reduces instabilities during the early training.

## I.2  GI-Adam as an Automated Warmup

In this section, we show that a bias correction is not required when the second moment is initialized with the gradients at initialization in GI-Adam. Therefore, employing a bias correction as in the original Adam algorithm in this case serves as an automated warmup given by $\eta_t = \eta_{\mathrm{trgt}} \sqrt{1 - \beta_2^t}$.

The moving average of the second moment is given by:

$$\boldsymbol{v}_t = (1 - \beta_2) \sum_{i=0}^{t-1} \beta_2^i \boldsymbol{g}_{t-i}^2 + \beta_2^t \boldsymbol{v}_0, \tag{19}$$

where $\boldsymbol{v}_0 = \boldsymbol{g}_0^2$. Following standard assumptions, we assume that the second moment of the gradient is constant during early training $\mathbb{E}[\boldsymbol{g}_t^2] = \sigma^2$. Taking the expectation of the above equation over the gradient distribution yields

$$\mathbb{E}[\boldsymbol{v}_t] = (1 - \beta_2) \sum_{i=0}^{t-1} \beta_2^i \mathbb{E}[\boldsymbol{g}_{t-i}^2] + \beta_2^t \mathbb{E}[\boldsymbol{v}_0]. \tag{20}$$

Simplifying the above equation, we have

$$\mathbb{E}[\boldsymbol{v}_t] = (1 - \beta_2) \sigma^2 \frac{1 - \beta_2^t}{1 - \beta_2} + \beta_2^t \sigma^2 = \sigma^2. \tag{21}$$

This result demonstrates that when the second moment is initialized with the gradients at initialization, it does not require bias correction, as the expected value of the second moment is equal to the constant $\sigma^2$. If we apply the usual bias correction on top of initializing the second moment with the gradients, we effectively downscale the second moment by a factor $\sqrt{1 - \beta_2^t}$. Assuming small enough $\epsilon$, this can be viewed as a multiplicative factor to the learning rate. As a result, GI-Adam is equivalent to having a natural warmup given by $\eta_t = \eta_{\mathrm{trgt}} \sqrt{1 - \beta_2^t}$.

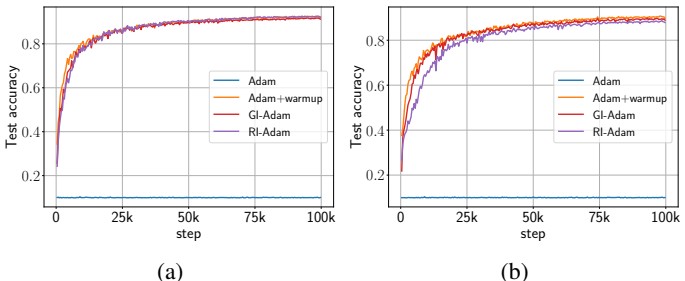

(a)                                    (b)

Figure 33: Comparison of test accuracy trajectories of WRNs trained with different Adam variants for two target different learning rates: (a) $\eta_{\text{trgt}} = 0.020480$, and (b) $\eta_{\text{trgt}} = 0.040960$. For Adam+warmup, the warmup duration is set to $T_{\text{wrm}} = 1024$.

### I.3 The Primary benefit of GI-Adam results from the magnitude of the second moment at initialization

To further assess if the primary cause of instability during early training is the large $\lambda^{P^{-1}H}$, we randomly initialize $v_0$ but with the same norm as the gradients at initialization. We refer to this as Randomly Initialized Adam (RI-Adam). Like GI-Adam, this also results in improved performance as shown in Figure 33.

### I.4 Warmup Mechanisms of Flat-Adam

Figure 34 shows the training trajectories of FCNs with different parameterizations trained with Flat-Adam. We observe that further removing bias correction for the first moment completely removes the initial reduction in the pre-conditioned sharpness.

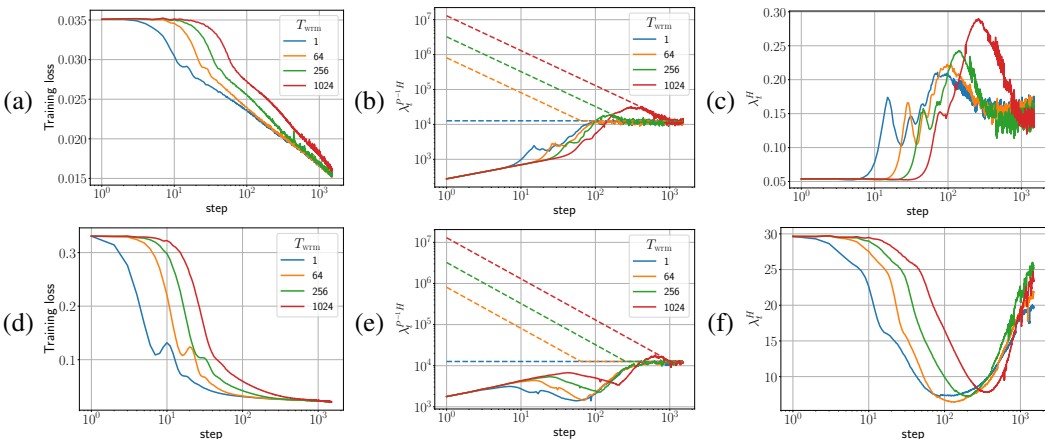

Figure 34: Training loss and sharpness trajectories of FCNs in (top) $\mu$P and (bottom) SP. The experimental setup is identical to Figure 2 but with Flat-Adam instead of standard Adam.

