# OpenReview forum: "Why Warmup the Learning Rate? Underlying Mechanisms and Improvements"
_NeurIPS.cc/2024/Conference — NeurIPS 2024 poster_

### Official Review · Reviewer_kKLj · 2024-07-06

**Soundness:** 3
**Presentation:** 3
**Contribution:** 3
**Rating:** 7
**Confidence:** 4

**Summary:**

The paper analyses different aspects of warmup in the gradient based training focusing on SGD, ADAM and their variants under two types of parameterization. It shows through mostly empirical analysis that warmup facilitates training at higher learning rates and stabilizes the training dynamics by keeping it away from (what they call) divergence boundary (that results in failure). Furthermore, it suggests several improvements for hyperparameter initialization that shorten the training process and improves generalization.

**Strengths:**

Choosing the learning rate  is critical for training large models. The paper proposes a nice analysis of the warmup procedure that agrees with some of the previous observations. And it suggests useful tips for practitioners.

**Weaknesses:**

The figures are not quite clear, especially because the captions do not describe the figures well enough.
The observations are made mostly from empirical study.

**Questions:**

The study is done on convnets and resnets applied to images. Training LLMs is a more complex task. Would these ideas apply to attention based architectures? Or more widely, do the mechanisms described in the paper depend on the architecture?

**Limitations:**

Yes

---

> ### Author Rebuttal · Authors · 2024-08-06
>
> We thank the reviewer for taking the time to review our paper and for their encouraging comments. Below are our responses for the questions and comments.
>
> > The figures are not quite clear...
>
> We noticed some errors in the Figure 3 caption in the submission and have revised it as follows:
> ''Test accuracy heatmaps of WRNs trained on CIFAR-10 using different and parameterizations loss functions using SGD: (a) $\mu$P and MSE loss, (b) $\mu$P and cross-entropy loss, (c) SP and MSE loss, and (d) SP and cross-entropy loss. Empty cells correspond to training divergences. Similar phase diagrams are generically observed for different architectures and datasets, as shown in Appendix F.''
>
> > The observations are made mostly from empirical study,,
>
> We can use a toy model to understand the two warmup mechanisms. Following Ref. [1], we can understand the self-stabilization mechanism through a toy model resulting from a third-order approximation of the loss function.
>
> Consider a loss function $L(\theta)$ with parameters $\theta$. Let $\lambda^H_t$ and $u$ denote the sharpness and its corresponding eigenvector. The model assumes that the top eigenvector $u$ changes slowly through training and can be treated as constant. Next, consider a cubic approximation of the dynamics along a reference point $\theta^*$. The dynamics along the projection $x_t:= u^T (\theta_t - \theta^*)$ is given by two coupled non-linear equations:
>
> \begin{align}
>         & x_{t+1} = (1 - \eta_t \lambda_t^H)x_t,\\
>         & \lambda_{t+1}^H = \lambda_t^H + \eta_t (\alpha - \beta x^2_t),
> \end{align}
>
> where $\alpha := - \nabla \lambda^H \cdot \nabla L(\theta)$ quantifies the instantaneous change in sharpness and $\beta:= \|\nabla \lambda^H  \|^2$ controls to the non-linear change in sharpness. Ref. [1] considered a constant learning rate $\eta$ and $\alpha > 0$. Here, in contrast, we consider a time-dependent learning rate and allow $\alpha$ to attain both positive and negative values.
>
> In this model, an instability arises when $\eta_t \lambda_t^H  > 2$. During instability, $x_t$ continues to increase until the higher order term in the sharpness update equation causes a significant decrease in sharpness. Once the sharpness has decreased sufficiently, the stability is restored ($\eta_t \lambda_t^2 < 2$), and training continues.
>
> Next, we consider the two natural sharpness evolution scenarios considered in our work:
>
>  1. **Natural progressive sharpening ($\alpha > 0$):** The combined effect of naturally increasing sharpness ($\alpha > 0$) and the increasing learning rate from warmup leads to instability ($\eta_t \lambda_t^H > 2$). Resultantly, $x_t$ increases until the higher order term in the sharpness update cause a decrease in sharpness ($x_t^2 > \frac{\alpha}{\beta}$). Once the sharpness has decreased appreciably so that $\eta_t \lambda^H_t < 2$ , stability is restored and the training continues. As training proceeds, both progressive sharpening and increasing learning rate cause instability, resulting in a persistent catapult cycle characterized by $\eta_t \lambda_t^H \approx 2$.
>
> 2. **Natural Sharpness Reduction ($\alpha < 0$):**
> In this case, sharpness is naturally decreasing during training ($\alpha < 0$). If the learning rate is increased quick enough relative to decreasing sharpness, an instability occurs $(\eta_t \lambda_t^H > 2)$. The increase in $x_t$ causes a more pronounced decrease in sharpness than it would have occurred naturally, restoring instability. To exceed the instability threshold again, the learning rate must significantly increase to account for the decreased sharpness. This results in one or more separated catapults.
>
> We will include this toy model in the updated version of our paper, which will complement our experimental results in Section 4.
>
> [1] `Self-Stabilization: The Implicit Bias of Gradient Descent at the Edge of Stability', Alex Damian, Eshaan Nichani, Jason D. Lee, ICLR 2023
>
> > The study is done on convnets and resnets applied to images...
>
> We have extended our experiments to include Transformers trained on language tasks and found that our results also apply to attention-based methods. These results are detailed in the global response section.

---

> > ### Comment · Reviewer_kKLj · 2024-08-09
> >
> > Thank you for the reply. I think that these additions to the paper will make it even stronger.

---

### Official Review · Reviewer_YVuK · 2024-07-11

**Soundness:** 3
**Presentation:** 3
**Contribution:** 3
**Rating:** 7
**Confidence:** 4

**Summary:**

This paper studies the mechanisms of the warmup technique. The authors experimentally demonstrate that the primary benefit of warmup is its ability to enable the network to handle larger learning rates.

**Strengths:**

Warmup is an essential trick for training modern deep neural networks, and understanding its role is a critical and open issue.
This paper makes a significant contribution to this exploration.
From the perspective of training stability, the author highlights that the use of warmup enables network training to utilize larger learning rate.
Additionally, the author notes that different initializations correspond to different stability regimes at the beginning of training: sharpness reduction and progressive sharpening.
Warmup is particularly important for maintaining training stability in the sharpness reduction regime.

**Weaknesses:**

- The experiments are primarily conducted on ResNet models on CIFAR, where warmup is not essential.
However, for Transformer-based models, warmup appears to be indispensable, particularly in applications such as language model pretraining and Vision Transformer (ViT) training.

- The observations in this paper are all based on experimental results. The findings would be more convincing if they could be theoretically validated in some settings (even in toy settings).

**Questions:**

- It is natural and insightful that extending warmup's time can tolerate the use of larger LR, but can the authors further clearly and quantitatively indicate this relationship?

- Does the measure of sharpness, $\lambda_{\max}(H)$, apply to randomized algorithms? For example, for GD, the stability condition is typically $\lambda_{\max}(H)\leq 2/\eta$ ; however, for SGD, $\lambda_{\max}(H)$ may no longer be suitable sharpness measure about training stability, and it might be $||H||_{\rm F}$ [1]. As the authors show in Figure 1(d) for GD, the results fully support the author's claim; however, in Figure 11(d) for SGD, the results do not completely align with the author's claim.

- A closely related work [2] should be discussed. In Section 5 and Fig 3 in [2], the authors also discussed how initialization, warmup, and SGD noise influence progressive sharpening or sharpness reduction.


[1] Wu et al. The alignment property of SGD noise and how it helps select flat minima: A stability analysis. (NeurIPS 2022)

[2] Ziyin et al. Loss Symmetry and Noise Equilibrium of Stochastic Gradient Descent. (2024)

**Limitations:**

Please refer to ``Weaknesses''.

---

> ### Author Rebuttal · Authors · 2024-08-06
>
> We thank the reviewer for reviewing our paper and they encouraging feedback.
>
> > The experiments are primarily ....
>
> We have extended our experiments to include Transformers trained on language modeling tasks and found that our results extend to these models. The results are detailed in the global response. Furthermore, we would like to draw your attention to Appx F of our submission, where we show phase diagrams for CIFAR-100 and TinyImageNet.
>
> > observations in this paper are all based on..
>
> We can use a toy model to understand the two warmup mechanisms. Following Ref. [3], we can understand the self-stabilization mechanism through a toy model resulting from a third-order approximation of the loss function.
>
> Consider a loss function $L(\theta)$ with parameters $\theta$. Let $\lambda^H_t$ and $u$ denote the sharpness and top eigenvector. The model assumes that the top eigenvector $u$ changes slowly through training and can be treated as constant. Next, consider a cubic approximation of the dynamics along a reference point $\theta^*$. The dynamics along the projection $x_t:= u^T (\theta_t - \theta^*)$ is given by two coupled equations
>
> $$ x_{t+1} = (1 - \eta_t \lambda_t^H)x_t, $$   $$ \lambda_{t+1}^H = \lambda_t^H + \eta_t (\alpha - \beta x^2_t), $$
>
> where $\alpha := - \nabla \lambda^H \cdot \nabla L(\theta)$ quantifies the instantaneous change in sharpness and $\beta:= \|\nabla \lambda^H  \|^2$ controls to the non-linear change. Ref. [3] considered a constant learning rate $\eta$ and $\alpha > 0$. Here, in contrast we consider a time-dependent learning rate and allow $\alpha$ to attain both positive and negative values.
>
> In this model, an instability arises when $\eta_t \lambda_t^H  > 2$. During instability, $x_t$ continues to grow until the higher order term in the sharpness equation causes a significant decrease in sharpness. Once the sharpness has decreased sufficiently, the stability is restored ($\eta_t \lambda_t^2 < 2$), and training continues.
>
> Next, we consider the two natural sharpness evolution scenarios considered in our work:
>
> 1. **Natural progressive sharpening ($\alpha > 0$):** The combined effect of naturally increasing sharpness ($\alpha > 0$) and the increasing learning rate from warmup leads to instability ($\eta_t \lambda_t^H > 2$). Resultantly, $x_t$ increases until the higher order term in the sharpness update cause a decrease in sharpness ($x_t^2 > \frac{\alpha}{\beta}$). Once the sharpness has decreased appreciably so that $\eta_t \lambda^H_t < 2$, stability is restored and the training continues. As training proceeds, progressive sharpening and increasing learning rate cause instability, resulting in a persistent catapult cycle characterized by $\eta_t \lambda_t^H \approx 2$.
>
> 2. **Natural Sharpness Reduction ($\alpha < 0$):** In this case, sharpness is naturally decreasing during training ($\alpha < 0$). If the learning rate is increased quick enough relative to decreasing sharpness, an instability occurs $(\eta_t \lambda_t^H > 2)$. The increase in $x_t$ causes a more pronounced decrease in sharpness than it would have occurred naturally, restoring instability. To exceed the instability threshold again, the learning rate must significantly increase to account for the decreased sharpness. This results in one or more separated catapults.
>
> We will include this toy model in the updated version of our paper, which will complement our experimental results in Section 4.
>
> [3] Self-Stabilization: The Implicit Bias of Gradient Descent at the Edge of Stability, ICLR 2023
>
> > It is natural and insightful that ....
>
> We have already qualitatively demonstrated that increasing warmup duration facilitates training at higher target learning rates. Figs 1 and 2 illustrate that increasing the warmup duration results in smaller loss catapults, indicating improved stability at higher learning rates. This observation can also be argued through the toy model presented in the prior response. Figs 3 and 4 specifically show how the maximum target learning rate scales with warmup duration. To further address the reviewer's request for a more quantitative analysis, we can provide fitted curves to the maximum learning rate - warmup duration trends in the final version of the manuscript.
>
> > Does the measure of sharpness...
>
> Our extensive empirical analysis shows that even for randomized algorithms like SGD, the relevant stability measure is $\lambda_{max}(H)$, not $||H||_F$. However, it is true that the stability threshold can change significantly for small batch sizes [1].
> Prior empirical work [4] has indeed shown that $\lambda^H$ oscillates at a lower threshold than $\frac{2}{\eta}$ at late training times. Nevertheless, our results in Appendix E show that similar warmup mechanisms are observed for SGD, albeit saturating at a lower threshold (Fig 10 & 13).
>
> The discrepancy in Fig 11(d) can be primarily attributed to loss function rather than the use of SGD. Fig 8(d) shows an experiment with the same batch size but with MSE loss. At late training times, $\lambda^H$ oscillates slightly below $\frac{2}{\eta}$, suggesting a minimal deviation from GD. In comparison, for cross-entropy loss (Fig 11d), we observe that (i) sharpness oscillates slightly above $\frac{2}{\eta}$ during training and (ii) sharpness dramatically decreases towards the end. These phenomena align with the findings from prior studies.
> Ref. [4] showed that sharpness decreases at the end of training for cross-entropy loss. Ref. [5] observed that for cross-entropy, the loss starts to catapult around $\eta \approx \frac{4}{\lambda^H}$.
>
> [4] Gradient descent on neural networks typically occurs at the edge of stability, ICLR, 2021
>
> [5] Phase diagram of early training dynamics in deep neural networks, NeurIPS 2023.
>
> > A closely related work [2] ....
>
> We thank the reviewer for bringing Ref. [2] to our attention. We will incorporate a discussion of this paper in the related works section of our updated manuscript.

---

> > ### Comment · Reviewer_YVuK · 2024-08-09
> >
> > Many thanks to the authors for the detailed response. I feel that this paper provides great insights into Warmup. I have raised my score.

---

### Official Review · Reviewer_LAU1 · 2024-07-12

**Soundness:** 3
**Presentation:** 3
**Contribution:** 3
**Rating:** 6
**Confidence:** 4

**Summary:**

The authors explain the mechanisms of the warmup technique showing that with warm up the loss of NN will go to a flatter space than direct optimization. Further, based on analysis, the authors propose a new optimization algorithm called GI-Adam.

**Strengths:**

1. The authors explain why the warm-up technique can help networks converge better.

2. With the analysis, the authors show that the initialization of Adam is not "correct". Thus, the authors proposed a new initialization of Adam called GI-Adam.

**Weaknesses:**

1. The conclusions are from FCNs and WideResnet, which can be trained well without warm-up. Does the conclusion still hold for some "hard" models and datasets (e.g., Transformer)?

2. GI-Adam is used in [1].

[1] Zhang, Yushun, Congliang Chen, Naichen Shi, Ruoyu Sun, and Zhi-Quan Luo. "Adam can converge without any modification on update rules." Advances in neural information processing systems 35 (2022): 28386-28399.

**Questions:**

See Weaknesses.

---

> ### Author Rebuttal · Authors · 2024-08-06
>
> We thank the reviewer for their time and effort for reviewing our paper and providing comments.
>
> > The conclusions are from FCNs and WideResnet, which can be trained well without warm-up. Does the conclusion still hold for some "hard" models and datasets (e.g., Transformer)?
>
> We have extended our experiments to include Transformer models trained on language modeling tasks with SGD and Adam. Our findings demonstrate that the conclusions drawn from FCNs and WideResNets generalize to Transformers trained on language modeling tasks. The additional results are presented in the global response.
>
> > GI-Adam is used in [1].
>
> We have reviewed Ref. [1] and respectfully disagree. Below we would like to take the opportunity to clarify the distinction between the Adam initialization used in Ref. [1] and our proposed GI-Adam. In their theoretical analysis, Ref. [1] initializes both first and second moments using gradients at initialization as a replacement for bias correction (Algorithm 1). This approach is primarily used for simplifying their theoretical analysis and does not include empirical evaluation of their proposed initialization.
>
> In comparison, GI-Adam initializes the second-moment using gradients ($v_0 = g_0^2$), while initializing the momentum to zero ($m_0 = 0$) and keeping the bias corrections. As shown in the derivation below, under standard assumptions for deriving the bias correction, initializing the second moment with the gradients at initialization does not require bias correction (also mentioned in Ref. [1]). Hence, for small $\epsilon$, the bias correction on top of setting $v = g_0^2$ can be viewed as a multiplicative factor to the learning rate. As a result, GI-Adam is equivalent to having natural warmup given by $\eta_t = \eta_{\text{trgt}} \sqrt{1 - \beta_2^t}$.
>
> Furthermore, the referenced study has not performed any empirical analysis of their Adam initialization and it appears to be used as a replacement for bias correction for simplifying the mathematical analysis. In comparison, we show that GI-Adam has a smaller pre-conditioned sharpness at initialization compared to Adam and requires less warmup.
>
> These distinctions highlight that GI-Adam is a novel approach, different from the Adam initialization in Ref. [1]. Nevertheless, we acknowledge the relevance of the reference and will cite it in the related works section of our updated manuscript.
>
>
> **Derivation:** The moving average of the second moment is given by:
>
> \begin{align}
> v_t = (1 - \beta_2) \sum_{i=0}^{t-1}    \beta_2^i g_{t-i}^2 + \beta_2^t v_0,
> \end{align}
>
> where $v_0 = g_0^2$. Following standard assumptions, we assume that the second moment of the gradient is constant during early training $\mathbb{E}[g_{t}^2] = \sigma^2$. Taking the expectation of the above equation over the gradient distribution yields
>
> \begin{align}
>     \mathbb{E} [ v_t ] = (1 - \beta_2) \sum_{i=0}^{t-1} \beta_2^i \mathbb{E} [ g_{t-i}^2] + \beta_2^t \mathbb{E}[ v_0].
> \end{align}
>
> Simplifying the above equation, we have
>
> \begin{align}
>     \mathbb{E}[v_t] = (1 - \beta_2) \sigma^2 \frac{1 - \beta_2^t}{1 - \beta_2} + \beta_2^t \sigma^2 = \sigma^2.
> \end{align}
>
> Therefore, initializing the second moment with gradients at initialization does not require the corresponding bias correction.

---

> > ### Comment · Reviewer_LAU1 · 2024-08-09
> >
> > Thanks for the authors' response. I have no further questions and raised my score.

---

### Official Review · Reviewer_6XDG · 2024-07-13

**Soundness:** 3
**Presentation:** 3
**Contribution:** 3
**Rating:** 6
**Confidence:** 4

**Summary:**

The paper examines the learning rate warmup technique from the perspective of its influence on the evolution of loss sharpness for different optimizers (GD, SGD(-M), Adam) and network parametrizations (Maximal Update Parameterization – $\mu$P and Standard Parametrization – SP). The authors demonstrate that the warmup allows the network to tolerate larger learning rates by gradually reducing sharpness through a series of loss catapults and self-stabilizations. Experiments with different network parameterizations reveal how warmup influences training in progressive sharpening and sharpness reduction regimes and the lower importance of warmup for training $\mu$P parameterized networks. Based on the empirical analysis, the paper proposes two practical training heuristics: (1) initializing the learning rate at an estimated critical value to eliminate unnecessary warmup steps and (2) introducing the GI-Adam optimizer, which initializes Adam's second moment with a squared gradient.

**Strengths:**

1. The paper provides solid experiments demonstrating that the gradual self-stabilization mechanism induced by the warmup is observed for different network parameterizations and optimizers.
2. The paper discusses the specifics of the warmup effect on networks in different parameterizations and confirms that networks in $\mu$P parametrization benefit less from it.
3. The paper explains why warmup may be unstable for Adam and proposes a simple heuristic on how to deal with this instability. This heuristic potentially may be useful in practice.
4. The paper includes an extensive study of warmup hyperparameters (warmup length and maximal learning rate) and suggests how to choose their optimal values. I specifically like the Persistent Catapult Warmup idea from the appendix and think it is promising.
5. The paper is clearly written and easy to follow.

**Weaknesses:**

My main concerns are related to the level of novelty of the empirical analysis of the warmup and the effectiveness of the proposed practical modifications:
1. The novelty and significance of the empirical analysis in the first part of the paper seem limited. The study heavily relies on two previous works. Gilmer et al., 2021 (https://arxiv.org/pdf/2110.04369) investigate training instabilities from the perspective of sharpness, including a very similar analysis on how warmup decreases the sharpness and a discussion on how starting training from a flat initialization makes warmup much less important. Karla et al., 2023 (https://arxiv.org/pdf/2311.02076) demonstrate that $\mu$P and SP exhibit different natural evolutions of sharpness. The paper combines these two perspectives, however, I am not sure if this combination leads to new insights. The warmup works similarly in both progressive sharpening and sharpness reduction regimes. The paper points out the instability in Adam warmup and the lower importance of warmup for $\mu$P parametrized models (similar to the discussion in Gilmer et al., 2021), but both of these insights are not related to the different sharpness evolution regimes.
2. The authors claim that starting warmup from the critical learning rate $\eta_c$ is an effective strategy. However, this claim is not obvious, and adding empirical evaluation for it would improve the paper. For example, the experiment varying starting warmup learning rate can be added to demonstrate that starting from $\eta_c$ value results in a shorter or more stable warmup than for lower and higher values.
3. Test accuracy heatmaps are not provided for the experiments on the initial learning rate selection for warmup. Hence, it is not clear whether the proposed strategy results in high-quality solutions. Moreover, high values of $T_{\text{save}}$ are observed for the small $\eta_{\text{trgt}}$ and large $T_{\text{wrm}}$ (Figure 5b), but this configuration is clearly suboptimal since much shorter warmups work well for the low learning rates. For the most practically interesting hyperparameter regions associated with the shortest effective warmup for each learning rate, $T_{\text{save}}$ is negligible.
4. The maximum test accuracy achieved by GI-Adam appears indistinguishable from that of baseline Adam in most cases, so it is difficult to say whether the GI-Adam improves the training. Moreover, adding standard deviations to this comparison seems important due to the noisy behavior of the test accuracy. Also, GI-Adam is not the first Adam modification that increases stability at the beginning of training (see, e.g., RAdam from https://arxiv.org/pdf/1908.03265v4). A more accurate discussion of such methods and comparison with them would benefit the paper.
5. The paper lacks experiments with Transformer architecture, while it is a primary case of using Adam optimizer with warmup.

Minor comments
1. Line 64: the idea that warmup is unnecessary if training is stable with the chosen learning rate is obvious and widely used in practice.
2. Lines 185-186: I would not say that $\mu$P does not benefit from warmup at all. As shown in Figure 3a, a longer warmup in the case of MSE training extends the range of converging learning rates.

**Questions:**

I would kindly ask the authors to address the main concerns from the Weaknesses section and focus on the following questions:
1. Could you please summarize the main novel insights of the empirical analysis part of the paper compared to Gilmer et al., 2021 and Karla et al., 2023, and explain why analyzing the warmup behavior in different sharpness evolution regimes is important?
2. Could you please provide any experiments demonstrating that starting warmup from the critical learning rate $\eta_c$ is an effective strategy?
3. What is the test accuracy with optimal warmup hyperparameters for baseline Adam, Adam with $\eta_{\text{init}}=\eta_{c}$ and GI-Adam? Is there any statistically significant difference between them?
4. Is the same self-stabilization mechanism observed when training Transformers?

Additional minor questions:
1. Why do you use different target learning rates for $\mu$P and SP in Figure 1? It is a bit confusing since the target learning rate and parameterization are changed between the two experiments at the same time, and it is not clear which change results in which effect. At the same time, Figure 2 uses an identical learning rate for both initializations.
2. How do you measure the loss value for initial learning rate selection and the squared gradient for GI-Adam in the stochastic variants of the algorithms? Do you use a single batch or estimate these values over multiple batches? Using a single batch may result in higher variance in the estimates, which could be undesirable. On the other hand, estimating over several batches would incur additional computational costs.

There also exist several related works which the authors may find interesting:
* Lobacheva et al., 2021 (https://arxiv.org/pdf/2106.15739) report an effect similar to warmup self-stabilization when training scale-invariant networks with weight decay and a constant learning rate. The decreasing weight norm increases the effective learning rate, which eventually leads to training instability and catapults. This periodic behavior allows the network to achieve flatter optima with higher test accuracy after several cycles.
* A different cyclical behavior, the Slingshot effect, is observed in adaptive optimizers like Adam, as shown by Thilak et al., 2024 (https://openreview.net/pdf?id=OZbn8ULouY). This effect occurs in the terminal phase of training and involves a rapid growth of the last layer norm before catapulting, followed by an improvement in test performance.
* In your experiments, optimal test performance is achieved with large learning rates close to the convergence boundary. However, Kodryan et al., 2022 (https://arxiv.org/pdf/2209.03695) show that training networks with weight decay and learning rates larger than optimal usually lead not to divergence but to a noisy stabilization of test error. Moreover, Andriushchenko et al., 2022 (https://arxiv.org/pdf/2210.05337) demonstrate that further reducing the learning rate from these stabilized solutions results in the model learning sparser features and achieving better final test performance.

**Limitations:**

The authors adequately discuss the limitations of the paper in the conclusion.

---

> ### Author Rebuttal · Authors · 2024-08-06
>
> We thank the reviewer for their time and effort in reviewing our paper and providing comments.
>
> > 1. Could you summarize the main novel insights ...
>
> While Gilmer et. al. 2021 is a key reference that our paper builds on, there are several novel insights in the empirical analysis part of our paper. First, our work emphasizes qualitatively distinct phenomena that arise at early training times, depending on whether the network starts in progressive sharpening or sharpness reduction phases. While Kalra et. al. 2023 also discuss sharpness reduction and progressive sharpening at early training times, the implications for learning rate warmup are not discussed. Our work combines ideas from both papers with novel empirical analysis to demonstrate for the first time the two separate underlying regimes of operation for learning rate warmup.
>
> A key result of our analysis is that models that experience sharpness reduction require more warmup compared to models that experience progressive sharpening at initialization.
> This result forms the basis for examining Adam. Models trained with Adam only experience a reduction in pre-conditioned sharpness, regardless of natural sharpness evolution. It thus becomes evident why Adam generally requires warmup to perform well. This early reduction also suggests the possibility of flatter initializations for Adam, like GI-Adam, which we propose. Without a characterization of underlying warmup mechanisms in terms of early-time sharpness evolution, it is not obvious why Adam would generally require warmup.
> Therefore, the different sharpness evolution regimes are indeed related to the instability in Adam warmup.
>
> As for $\mu$P, while it has been demonstrated in Kalra et al., 2023 that $\mu$P is a flat initialization and one can speculate that it may not require warmup, our results show that models in $\mu$P can still benefit from warmup (Fig 3), as also pointed by the reviewer.
>
> Moreover, our analysis in Sec 5 disentangles the role of $T_{wrm}$ and $\eta_{trgt}$, which has not been performed in prior work. These results reveal that the final performance primarily depends on $\eta_{trgt}$, with longer warmup durations mainly helping to avoid the convergent-divergent (failure) boundary. This also leads us to the point that warmup has another advantage: it makes learning rate tuning more robust. This point was not, to our knowledge, made in prior work.
>
> We have also introduced persistent catapult warmup and shown some encouraging preliminary experiments in App. C. We intend to move these results to the main text.
>
> > 2. Could you provide experiments ...
>
> We have conducted experiments to demonstrate the effectiveness of starting warmup from the critical learning rate $\eta_c$. These results are shown in Fig 4 of the attachment to the global response. When training WRNs on CIFAR-10 using Adam, setting the initial learning rate to $\eta_c$ (referred to as Adam-save in these results) yields solutions of similar quality to Adam. In this case, we save 576 training steps. In the updated manuscript, we will include test accuracy heatmaps for initial learning rate selection to further illustrate this point.
>
> As described in Sec 6.1, the total number of steps saved is $T_{save} = T_{wrm} \eta_c / \eta_{trgt}$. For flat initializations, we have observed that $\eta_c$ is close to the optimal target learning rates. This means that for the optimal target learning rates, flat initializations can save the full warmup time, $T_{save} \approx T_{wrm}$. The benefit is less significant for large initializations where $\eta_c$ is small. Fig 22 in App H shows heatmaps of $T_{save}$ for WRNs in $\mu$P, where $\eta_c$ is close to the maximum trainable learning rate. In these cases, we observe that $T_{save} \approx T_{wrm}$ for most learning rates and warmup durations.
>
> Our analysis in Sec 4 demonstrates that before the `collision,' warmup does not actively reduce sharpness. Therefore, setting $\eta_{init} = \eta_c$ should not be expected to negatively impact the training dynamics. Furthermore, a priori, we cannot predict which part of the heatmap our chosen hyperparameters correspond to. Thus, adopting this strategy of starting from $\eta_c$ does not negatively impact training and potentially saves a significant number of training steps in favorable cases.
>
> > 3. What is the test accuracy with ...
>
> We have consistently observed that for small warmup durations, GI-Adam outperforms Adam. As shown in Table of Fig 4 of the attachment, when training WRNs on CIFAR-10, GI-Adam improves test accuracy by $0.5\%$ over Adam without warmup. This improvement is greater than the standard deviation in test accuracy across different initializations, indicating statistical significance. Moreover, in our language modeling experiments (detailed in the global response), we observed a loss improvement of $0.1$ with GI-Adam. For long warmups, we do not observe a significant advantage of GI-Adam over Adam. This is perhaps not surprising as GI-Adam performs a natural warmup, as described in the global response.
>
> Another benefit of GI-Adam is that it widens the range of optimal learning rates by pushing the failure boundary further, as demonstrated in Fig 4(b) of our submission. This makes $\eta_{trgt}$ easier to tune. Note that, a apriori, it is hard to predict where the experimental hyperparameters lie in the heatmap. Thus, it is always beneficial to use GI-Adam as it does not degrade performance, and yet it is a minor modification to Adam.
>
> We have also included Radam in our experiments for reference. GI-Adam performs similarly to Radam while being a significantly simpler modification to Adam.
>
> > 4. Is the same self-stabilization..
>
> We have extended our experiments to include Transformers trained on language datasets with SGD and Adam. Our results readily extend to this setting. These results are detailed in the global response.
>
> We provide replies to the minor questions in the comment below.

---

> ### Author Response · Authors · 2024-08-06
> **Responses to minor questions by Reviewer 6XDG**
>
> Here, we provide replies to minor questions by Reviewer 6XDG.
>
> > Why do you use different target learning rates for ...
>
> The different target learning rates for $\mu$P and SP in Figure 1 are chosen deliberately because we need to satisfy two requirements: (i) $\eta_{trgt}$ cannot be so large that training fails, and (ii) $\eta_{trgt}$ needs to be large enough so that warmup has a non-trivial effect. (i) and (ii) together give distinct viable values of $\eta_{trgt}$ for SP vs. $\mu$P.
>
> In comparison, we found that Adam's learning rates are relatively stable across parameterizations, which is why we used the same learning rate for both initializations in Figure 2 of the submission.
>
> > How do you measure the loss value...
>
> For both initial learning rate selection and squared gradient for GI-Adam we use a single batch for computations. Nevertheless, we did not observe any performance degradation for commonly used batch sizes as shown in Figure 6 of the PDF attached to the global response.
>
> For the initial learning rate estimation, small errors in estimating $\eta_c$ are not expected to impact training, as small initial loss spikes have minimal impact on the overall dynamics.
> For most models used in practice, $\eta_{\text{max}}$ is at least $4-8$ times larger than $\eta_c$ [1] and hence small errors in estimating $\eta_c$ would still be in the catapult phase $\eta_c < \eta < \eta_{\text{max}}$. However, we do agree with the reviewer that for small enough batch sizes, this estimate can be error-prone and or incur additional computational costs. We will mention it in the limitations section.
>
> [1] Lewkowycz, A., Bahri, Y., Dyer, E., Sohl-Dickstein, J. and Gur-Ari, G., 2020. The large learning rate phase of deep learning: the catapult mechanism. arXiv preprint arXiv:2003.02218
>
> > There also exist several related works ...
>
> We thank the reviewer for bringing these prior works to our attention. We will discuss them in the related works section.

---

> > ### Comment · Reviewer_6XDG · 2024-08-11
> >
> > Thank you for the detailed response and additional experimental results!
> >
> > I am still a bit confused about the analysis of the two warmup regimes and the importance of the difference between them. Based on the paper and rebuttal, I would summarize the main results as follows:
> > * warmup influences training dynamics differently depending on the behavior of the sharpness at the beginning of training (progressive sharpening or sharpness reduction),
> > * warmup is more important for the sharpness reduction regime.
> >
> > Could you please provide some clarifications on the following concerns regarding these results:
> > 1. I fail to see significant differences in the warmup influence between the two regimes. In both of them, training with a high initial learning rate may lead to strong catapults in training, and warmup allows the network to experience smaller sequential catapults instead. More catapults can be observed in the progressive sharpening regime, but it is unclear to me why it is important.
> > 2. The importance of the warmup seems to be much more related to the difference between the sharpness of the initialization and the critical sharpness for the target learning rate than to the decreasing/increasing sharpness. The warmup may be crucial for training networks with progressive sharpening if we want to use learning rates higher than the critical threshold. At the same time, warmup is unnecessary for training networks with sharpness reduction with small enough learning rates. Why does increasing or decreasing sharpness define the importance of warmup and not just the difference between the sharpness at initialization and the critical sharpness for the target learning rate?
> > 3. In most experiments, the initial sharpness reduction quickly transitions to progressive sharpness during warmup steps, and all warmup catapults take place in the progressive sharpening regime (see Fig. 2 in the paper and Fig. 1,2 in the rebuttal). This observation makes the claim that warmup is more important for the sharpness reduction regime even more confusing.

---

> ### Author Response · Authors · 2024-08-12
>
> We thank the reviewer for their questions and comments. We hope other concerns regarding the initial learning rate, GI-Adam, and Transformers have been resolved.
>
>
> > The importance of the warmup seems to be much more related to the difference between the sharpness of the initialization and the critical sharpness for the target learning rate than to the decreasing/increasing sharpness. The warmup may be crucial for training networks with progressive sharpening if we want to use learning rates higher than the critical threshold. At the same time, warmup is unnecessary for training networks with sharpness reduction with small enough learning rates. Why does increasing or decreasing sharpness define the importance of warmup and not just the difference between the sharpness at initialization and the critical sharpness for the target learning rate?
>
> We agree with the reviewer that the necessity of warmup is based on the initial sharpness relative to the target learning rate. In our paper, when we made the statement that sharpness reduction regimes necessitate more warmup, it was for the case where the target learning rate was large and close to the optimal value. For such fixed optimal target learning rates, we found that networks that start off in the sharpness reduction phase necessitate warmup more than networks that start off in the progressive sharpening regime. The reason for this is that empirically there is a direct correlation between the initial sharpness and whether one observes sharpness reduction or progressive sharpening. Sharpness reduction in early training implies that the initial sharpness is "large," which then implies that warmup will be more important. We will modify the wording of our paper to make this point more clear and avoid this confusion.
>
> We would like to further point out that the main utility of understanding which regime we are in is that it can provide a way of defining whether the sharpness is "large" or "small." This in turn can give a clear indication of whether there is a better choice of initialization. For example, if one starts with a given sharpness and observes sharpness reduction phenomena, then it indicates that there is naturally a flatter initialization that one can pick. Similarly, if one starts with that same sharpness but sees progressive sharpness phenomena, then it is unclear, perhaps even unlikely, that a flatter initialization can be found. This was precisely what led us to discover GI-Adam. If we had found that the usual Adam initialization has high preconditioned sharpness but starts off in a progressive sharpening phase, then it would have been less clear that there might be a better possibility.
>
> > In most experiments, the initial sharpness reduction quickly transitions to progressive sharpness during warmup steps, and all warmup catapults take place in the progressive sharpening regime (see Fig. 2 in the paper and Fig. 1,2 in the rebuttal). This observation makes the claim that warmup is more important for the sharpness reduction regime even more confusing.
>
> This argument does not take into account the convergence/failure boundary. If we start off in progressive sharpening, the network can tolerate relatively high warmup rates (i.e. small $T_\text{wrm}$) according to Fig. 4(a, b). But if we start off in the sharpness reduction regime, even if we crossover to progressive sharpening after $10-20$ steps, those first $10-20$ steps will severely limit the warmup rate: High warmup rates will cause the training to diverge/fail. Therefore this early-time training dynamics is crucial for setting a maximum speed limit on warmup.
>
> We note that it is possible that the deeper understanding developed here might lead to more sophisticated warmup schedules; one can imagine starting with a low warmup rate while the network is in the sharpness reduction phase and transitioning to a high warmup rate after some time.

---

> > ### Author Response · Authors · 2024-08-12
> >
> > > I fail to see significant differences in the warmup influence between the two regimes. In both of them, training with a high initial learning rate may lead to strong catapults in training, and warmup allows the network to experience smaller sequential catapults instead. More catapults can be observed in the progressive sharpening regime, but it is unclear to me why it is important.
> >
> > We believe the importance of understanding the two qualitatively distinct regimes can be summarized as follows:
> >
> > 1. Having as deep an understanding of the underlying dynamics as possible is intrinsically valuable. Such developments in understanding may be followed by unanticipated innovations and can inform future decisions about the design of algorithms, initializations, and architectures. We think these unanticipated developments are likely to be most important. Our understanding of the different regimes is already important for the practical innovations proposed in our work, as we explain below.
> >
> > 2.  As we mentioned in our previous reply, our proposal for GI-Adam was motivated entirely by understanding that the usual Adam has an unnecessarily large preconditioned sharpness, leaving it deep in the sharpness reduction phase and that there is a simple tweak to the initialization that can reduce the need for warmup. We note that this analysis also led us to the understanding that the explanations of the RAdam paper were incorrect and that the RAdam algorithm was unnecessarily complicated.
> >
> > 3. In our paper we have suggested the persistent catapult warmup schedule, for which we have developed some preliminary analysis and left a complete development for future work. The hyperparameters of this schedule, $\delta$ in Algorithm 3 in Appendix C, which specifies the amount of increase in loss that can be tolerated, depend crucially on whether the training is in the sharpness reduction regime or the progressive sharpening regime and the strength of the individual loss catapults.
> >
> >  4. Our suggestion for picking $\eta_{init} = \eta_c$ was originally motivated by analyzing the sharpness curves in the progressive sharpening case, and realizing how much time was being wasted. If we had not analyzed the early-time training dynamics carefully, this realization would have eluded us.
> >
> >  5. Our analysis shows clearly that the sharpness reduction regime is suboptimal because it requires a longer warmup duration for optimal target learning rates. This is an important observation because it demonstrates there are ways to achieve the same test accuracy while requiring fewer warmup steps.
> >
> >
> >  6. The sharpness reduction regime shows that catapult / self-stabilization effects are not always the only mechanism by which warmup works. Another mechanism is via the natural sharpness reduction effect, where sharpness naturally reduces on its own even without a catapult. Therefore warmup can allow the network to tolerate larger learning rates without ever inducing catapults. This is a novel point that was not mentioned in prior work to our knowledge.

---

> > > ### Comment · Reviewer_6XDG · 2024-08-13
> > >
> > > Thank you for the explanation, it is much more clear than the one in the initial paper submission. After reading other reviews and authors' responses, I am increasing my score to 6.
> > >
> > > The concerns from my initial review were related to the three main points: the importance of the analysis of two warmup regimes (weakness 1), the importance of the proposed practical tricks (weaknesses 3 and 4), and some technical questions about experiments (weaknesses 2-5). Below, I summarize how my evaluation of each of them changed after the rebuttal:
> > > 1. Two warmup regimes. It was my main concern, and the authors reasonably addressed it in the last response. Specifically, I find the discussion of the correlation between the sharpness level and its increasing/decreasing behavior at the initialization and point 6 on the difference between the two regimes very helpful. I would ask the authors to incorporate these ideas in the final version of the paper. I think adding a discussion on the correlation between the sharpness level and its behavior is especially important. Without it,  the conclusion about the connection between sharpness behavior and the importance of the warmup seems vague and unsupported.
> > > 2. Practical tricks. I still believe that shortening the warmup does not improve effectiveness significantly. The results of Gi-Adam are not better than alternatives (RAdam), but after the rebuttal, I think the method is simple, well-motivated, and works better than the Adam baseline. Overall, I think the proposed practical modifications are reasonable, even if they are not the strongest contributions of the paper.
> > > 3. Technical questions. The authors fully addressed all my technical questions and provided additional experiments.

---

> ### Author Response · Authors · 2024-08-13
>
> We thank the reviewer for the insightful discussion. We agree that adding a discussion on the sharpness level and its training behavior will significantly improve the paper. We will incorporate this discussion along with other insightful suggestions in the final manuscript.

---

### Official Review · Reviewer_ZUpa · 2024-07-17

**Soundness:** 2
**Presentation:** 3
**Contribution:** 2
**Rating:** 5
**Confidence:** 3

**Summary:**

This paper finds that the learning rate warmup allow the network to tolerate larger learning rates. It gradually reduces the sharpness and forces the model to leave poorly conditioned areas of the lossscape and move toward flatter regions which can tolerate larger learning rates.

**Strengths:**

1. This paper analyzes the mechanisms of warmup.
2. It also proposes a GI-Adam strategy, which is better than Adam.

**Weaknesses:**

1. Though the author claims to find that warmup allows for larger learning rates, it has been found in existing work such as "Gilmer, J., Ghorbani, B., Garg, A., Kudugunta, S., Neyshabur, B., Cardoze, D., Dahl, G.E., Nado, Z. and Firat, O., 2022, March. A loss curvature perspective on training instabilities of deep learning models. In International Conference on Learning Representations."  Further elaboration on the difference and novelty will make the paper more convincing.

2. They claimed to find "wasted time can be saved by making use of the catapult mechanism", but it seems this has been revealed in "Lewkowycz, A., Bahri, Y., Dyer, E., Sohl-Dickstein, J. and Gur-Ari, G., 2020. The large learning rate phase of deep learning: the catapult mechanism. arXiv preprint arXiv:2003.02218."

Not only the above two points, I can't well understand the novelty of this paper. I suggest further summarize the contribution part.

3. Could you give more explainations on the accuracy maps, such as Figures 3 and 4?

**Questions:**

I don't have questions.

**Limitations:**

Please refer to the Weaknesses

---

> ### Author Rebuttal · Authors · 2024-08-06
>
> > Though the author claims to find that warmup allows for larger learning rates, it has been found in existing work such as Gilmer 2022. Further elaboration on the difference and novelty will make the paper more convincing....I suggest further summarize the contribution part.
>
> Whille Gilmer 2022 indeed demonstrated that warmup allows for larger learning rates primarily for models trained with SGD, our work extends and differentiates itself in several key ways that were not studied in prior work:
>
> * **Warmup mechanisms of SGD:** Gilmer 2022 demonstrated that warmup gradually reduces sharpness to facilitate training at higher learning rates. However, we go even further and ''look under the hood" by showing that generically there are two distinct underlying mechanisms. We show that the interplay between natural sharpness evolution and increasing learning rate during warmup leads to either (1) a persistent catapult cycle in progressive sharpening cases, or (2) separated loss catapults in sharpness reduction cases.
>
> * **Warmup Mechanisms of Adam:** We provide a comprehensive analysis of Adam's underlying warmup mechanisms, especially during early training times, which was not analyzed in Ref. [1]. In particular, we find that the pre-conditioned sharpness, which determines stability, decreases during early training, *regardless of the natural evolution of sharpness.* This initial reduction in pre conditioned sharpness, before it eventually increases, suggests the existence of flatter initializations (wrt pre-conditioned sharpness) for Adam, which can enable training at higher learning rates from the start. Based on this insight, we propose a simple alternative initialization method called GI-Adam, which provides benefits similar to warmup and consistently improves over standard Adam by pushing the training failure boundary to higher target learning rates.
>
> * **The Phase Diagrams of Warmup:** The test accuracy heatmaps (Figures 3 and 4) demonstrate how the maximum target learning rate changes with warmup duration and disentangles the effect of warmup time and target learning rate. These phase diagrams reveal that the final performance primarily depends on the target learning rate, with longer warmup durations mainly helping to avoid the convergent-divergent (failure) boundary.
>
> * **Dual Advantage of Warmup:** Our results not only show that warmup improves model performance by allowing larger target learning rates, but also that warmup gives rise to a wider range of target learning rates that yield optimal results, which makes learning rate tuning more robust. This additional benefit of warmup depends on the phase diagram results and were not discussed in prior work.
>
> * **Initial Learning Rate Selection:** As the primary effect of warmup is to facilitate training at higher learning rates by annealing sharpness (or pre-conditioned sharpness for Adam), setting the initial learning rate to $\eta_c$ induces loss increase at initialization and thereby sharpness decrease right from initialization, saving warmup training steps. We provide a simple and practical method to pick $\eta_c$ based on the loss catapult mechanism.
>
> * **Persistent Catapult Warmup:** We have also introduced a potential parameter-free warmup strategy, which we refer to as 'persistent catapult warmup.' The central idea behind this strategy is to repeatedly induce catapults aimed to progressively reduce sharpness (or pre-conditioned sharpness), thereby facilitating training at higher learning rates without specifying warmup duration. We demonstrate encouraging preliminary experiments in Appendix C of the submission. We intend to move these results into the main text.
>
> > They claimed to find "wasted time can be saved by making use of the catapult mechanism", but it seems this has been revealed in
>
> We respectfully disagree with the reviewer. Lewkowycz et al 2020 introduced the catapult mechanism, however, it never suggested utilizing it as a way to estimate $\eta_c$ and then setting the initial learning rate in warmup to be equal to $\eta_c$. The development of this idea is unique to our work.
>
> > Could you give more explainations on the accuracy maps, such as Figures 3 and 4
>
> Figures 3 and 4 show the best test accuracy achieved during training as a function of target learning rate $\eta_{\text{trgt}}$ and warmup duration $T_{\text{wrm}}$. These phase diagrams of warmup also show the convergence-divergence boundary, indicated by empty cells, illustrating the interplay between warmup duration and the maximum trainable learning rate. These results reveal: (i) flat initializations such as $\mu$P benefit less from warmup, whereas large initializations such as SP may require long warmup to attain the optimal performance, (ii) The final performance primarily depends on the target learning rate and the improvement from increasing warmup duration comes from keeping training away from the convergent-divergent (failure) boundary, (iii) warmup makes learning rate tuning more robust, as mentioned above.

---

> ### Comment · Reviewer_ZUpa · 2024-08-14
>
> Thanks for the response. I have no further questions.

---

### Author Rebuttal · Authors · 2024-08-06

We thank the reviewers for their time and effort in reviewing our paper.  Based on reviewers feedback, we have added the following results:

1. **Language Modeling Experiments:** We have extended our experiments to include Transformer models trained on language modeling tasks using SGD and Adam. In these experiments, we trained 4 layer Pre-LN transformers on the WikiText-2 dataset. The results are shown in the PDF attached.

    * Figure 1 shows the warmup mechanisms for Transformers trained on Wikitext-2 using SGD. We observe that the initial sharpness of the Pre-LN Transformers in Standard Parameterization (SP) is surprisingly small (\~5) and exhibits progressive sharpening right from the onset. This low initial sharpness is due to the last LayerNorm, and removing this layer results in a large sharpness (\~200), and reveals early sharpness reduction behavior observed in FCNs and ResNets in SP.

    * Figure 2 shows the warmup mechanisms of Pre-LN Transformers in SP trained on Wikitext-2 using Adam. Consistent with our findings in image classification tasks, we observe a reduction in pre-conditioned sharpness during early training, even for flat initializations that show an increase in sharpness from the onset.

    * Figure 3 presents the phase diagrams of warmup for Transformers trained on Wikitext-2 using both Adam and our proposed GI-Adam. In line with our image classification results, we observe:

        1. The final performance is primarily determined by the target learning rate and increasing the warmup duration keeps training further away from the divergence (failure) boundary.

        2. GI-Adam exhibits a wider range of target learning rates that achieve optimal performance, compared to standard Adam.

    These findings support the generalization of our conclusions to Transformers trained on language modeling tasks.

2. **Toy model for the two mechanisms of warmup:** We can use a toy model to understand the two warmup mechanisms. Following Ref. [1], we can analyze the self-stabilization mechanism through a model derived from a third-order approximation of the loss function.

    Consider a loss function $L(\theta)$ with parameters $\theta$. Let $\lambda^H_t$ and $u$ denote the sharpness and its corresponding eigenvector. The model assumes that the top eigenvector $u$ changes slowly through training and can be treated as constant. Next, consider a cubic approximation of the dynamics along a reference point $\theta^*$. The dynamics along the projection $x_t:= u^T (\theta_t - \theta^*)$ is given by two coupled non-linear equations: $$ x_{t+1} = (1 - \eta_t \lambda_t^H)x_t, $$ $$ \lambda_{t+1}^H = \lambda_t^H + \eta_t (\alpha - \beta x^2_t),$$
    where $\alpha := - \nabla \lambda^H \cdot \nabla L(\theta)$ quantifies the instantaneous change in sharpness and $\beta:= \|\nabla \lambda^H  \|^2$ controls to the non-linear change in sharpness. Ref. [1] considered a constant learning rate $\eta$ and $\alpha > 0$. Here, in contrast, we consider a time-dependent learning rate and allow $\alpha$ to attain both positive and negative values.

    In this model, an instability arises when $\eta_t \lambda_t^H  > 2$. During instability, $x_t$ continues to increase until the higher order term in the sharpness update equation causes a significant decrease in sharpness. Once the sharpness has decreased sufficiently, the stability is restored ($\eta_t \lambda_t^2 < 2$), and training continues.

    Next, we consider the two natural sharpness evolution scenarios:

    * **Natural progressive sharpening ($\alpha > 0$):** The combined effect of naturally increasing sharpness ($\alpha > 0$) and the increasing learning rate from warmup leads to instability ($\eta_t \lambda_t^H > 2$). Resultantly, $x_t$ increases until the higher order term in the sharpness update cause a decrease in sharpness ($x_t^2 > \frac{\alpha}{\beta}$). Once the sharpness has decreased appreciably so that $\eta_t \lambda^H_t < 2$, stability is restored and the training continues. As training proceeds, both progressive sharpening and increasing learning rate cause instability, resulting in a persistent catapult cycle characterized by $\eta_t \lambda_t^H \approx 2$.

    * **Natural Sharpness Reduction ($\alpha < 0$):**  In this case, sharpness is naturally decreasing during training ($\alpha < 0$). If the learning rate is increased quickly enough relative to decreasing sharpness, an instability occurs $(\eta_t \lambda_t^H > 2)$. The increase in $x_t$ causes a more pronounced decrease in sharpness than it would have occurred naturally, restoring instability. To exceed the instability threshold again, the learning rate must significantly increase to account for the decreased sharpness. This results in one or more separated catapults.

    We will include this analysis in the updated version of our paper, which will complement our empirical results in Section 4.

    [1] Self-Stabilization: The Implicit Bias of Gradient Descent at the Edge of Stability, ICLR 2023

3. **Improved understanding of GI-Adam:** Since the submission, we have improved our understanding of GI-Adam. Initializing the second moment with initial gradients eliminates the need for bias correction (we can provide a derivation). Hence, for small $\epsilon$, a bias correction, in addition to setting $v = g_0^2$, can be viewed as a learning rate multiplier. As a result, GI-Adam can be viewed as Adam with a natural warmup given by $\eta_t = \eta_{\text{trgt}} \sqrt{1 - \beta_2^t}$.

4. **Persistent Catapult Warmup:** We have also introduced a parameter-free warmup strategy `persistent catapult warmup.' The central idea behind this strategy is to repeatedly induce catapults aimed to progressively reduce sharpness (or pre-conditioned sharpness), thereby facilitating training at higher learning rates without specifying warmup duration. We demonstrate encouraging preliminary experiments in Appendix C of the submission.

---

### Decision · Program_Chairs · 2024-09-25

**Decision:**

Accept (poster)

**Comment:**

This paper provides a detailed empirical analysis of the warmup mechanism in neural network training, focusing on its role in improving training stability and enabling the use of larger learning rates. The authors also introduce the GI-Adam optimization method, which aims to enhance the initialization of the Adam optimizer. Strengths of the paper include the solid experimental validation of the warmup process, particularly its influence in different sharpness evolution regimes (as noted by 6XDG and kKLj), and the extension of these findings to Transformer models, which addressed concerns about the generality of the results (raised by LAU1 and YVuK). Additionally, the paper’s insights into the interplay between warmup duration and learning rate are appreciated for their potential practical applications in training neural networks (mentioned by kKLj and YVuK).

However, there are some weaknesses. ZUpa expressed concerns about the novelty of the paper, specifically the lack of a clear distinction from previous research, which was not fully alleviated by the authors’ response. Additionally, 6XDG pointed out that while the proposed GI-Adam method is reasonable, it does not offer significant improvements over existing alternatives like RAdam, which limits its practical impact.

While concerns remain regarding the novelty and the practical effectiveness of GI-Adam when compared to existing methods, I believe these issues can be adequately addressed if the manuscript clearly discusses these points in the main text. The numerical analysis of the warmup mechanism, focusing on different regimes during training, provides significant insights that are valuable to the NeurIPS community. Therefore, I recommend accepting this paper.